# Essential renormalisation group

A. Baldazzi[1], R. Ben Alì Zinati[2], K. Falls[1*]

**1** SISSA – International School for Advanced Studies & INFN, via Bonomea 265,
I-34136 Trieste, Italy
**2** Sorbonne Université & CNRS, Laboratoire de Physique Théorique de la Matière
Condensée, LPTMC, F-75005, Paris, France
* kfalls@sissa.it

June 11, 2021

## 1 Abstract

We propose a novel scheme for the exact renormalisation group motivated by
the desire of reducing the complexity of practical computations. The key idea
is to specify renormalisation conditions for all inessential couplings, leaving
us with the task of computing only the flow of the essential ones. To achieve
this aim, we utilise a renormalisation group equation for the effective aver-
age action which incorporates general non-linear field reparameterisations. A
prominent feature of the scheme is that, apart from the renormalisation of
the mass, the propagator evaluated at any constant value of the field main-
tains its unrenormalised form. Conceptually, the scheme provides a clearer
picture of renormalisation itself since the redundant, non-physical content is
automatically disregarded in favour of a description based only on quantities
that enter expressions for physical observables. To exemplify the scheme's
utility, we investigate the Wilson-Fisher fixed point in three dimensions at
order two in the derivative expansion. In this case, the scheme removes all
order $\partial^2$ operators apart from the canonical term. Further simplifications oc-
cur at higher orders in the derivative expansion. Although we concentrate on
a minimal scheme that reduces the complexity of computations, we propose
more general schemes where inessential couplings can be tuned to optimise a
given approximation. We further discuss the applicability of the scheme to a
broad range of physical theories.

---

# 1  Introduction

Our mathematical descriptions of natural phenomena contain redundant, superfluous information which is not present in Nature. This follows since, for any given problem, we always have the basic liberty to re-express the set of dynamical variables in terms of a new, perhaps simpler, set. In this respect, our mathematical models fall into equivalence classes, where two models are considered to be physically equivalent if they are related by a change of variables. Natural phenomena are therefore described by an equivalence class of effective theories rather than a specific model. However, in practice, in order to test our models against experiment, we would like to find those models that reduce the time and effort needed to compute a given physical observable.

The renormalisation group (RG) provides a framework to iteratively perform a change of variables with the purpose of describing physics at different length scales. This, in practice, translates into a flow in a space spanned by the couplings which parameterise all possible interactions between the physical degrees of freedom. However, due to the aforementioned redundancies, this *theory space* is divided into equivalence classes. As a consequence, we do not have to compute the flow of all coupling constants, but instead, we only need to compute the flow of the *essential coupling constants*, which are those eventually appearing in expressions for physical observables. The other coupling constants, known as *the inessential couplings*, can take quite arbitrary values since changing them amounts to moving within an equivalence class. It follows, therefore, that an inessential coupling is any coupling for which a change in its value can be reabsorbed by a change of variables. The prototypical example of an inessential coupling is the one related to a simple linear rescaling or renormalisation of the dynamical variables, namely, in a field-theoretic language, the wave-function renormalisation. Actually, it is this transformation that gives the renormalisation group its name. However, there is an infinite number of other inessential couplings related to more general, non-linear changes of variables. As we will show explicitly, one is free to specify the values of all inessential couplings instead of computing their flow. This freedom can then be exploited to simplify or otherwise optimise the calculation of physical quantities of interest. In addition, this has the advantage that we automatically disentangle the physical information from the unphysical redundant content encoded in the inessential couplings. Such possibility has been advocated independently by G. Jona-Lasinio [1] and by S. Weinberg [2]. Although a perturbative approach has been put forward in [3], so far, no concrete non-perturbative implementation based on general non-linear changes of variables has been realised.

The purpose of this paper is to arrive at a concrete scheme of this type, with the explicit aim of reducing the complexity of computations within the framework of K. Wilson's exact RG [4,5]. We shall refer to this concrete scheme as the *minimal essential scheme*. Essential schemes can be defined more generally as those for which we only compute the running of the essential couplings, having specified renormalisation conditions that determine the values of the inessential couplings as functions of the former.

To achieve our aim, in Section 2 we first develop the concept of field reparameterisations in quantum field theory (QFT). These changes of variables can be understood geometrically as local *frame transformations* on configuration space. After introducing the notation of a frame transformation for a classical field theory, we present a frame covariant formulation of QFT, where no particular frame is preferred a priori. In this way, it becomes manifest that observables are invariant under frame transformations. This leads to a precise definition of an inessential coupling and its conjugate *redundant operator*, whose identification is crucial to the concrete implementation of essential schemes. In the

rest of the paper, we combine this frame covariant formalism with a generalised version of the exact RG.

In the many years since K. Wilson first conceived of it, the exact RG, a.k.a. the non-perturbative functional renormalisation group has become a powerful technique that can be used to investigate a wide range of physical systems without relying on perturbation theory [6–12]. The fundamental idea consists of introducing a momentum space cutoff at the scale $k$ into the theory which allows the high momentum degrees of freedom $p^2 > k^2$ to be integrated out to obtain an effective action for the low momentum degrees of freedom. Its modern formulation is based on an exact flow equation [13,14] for the Effective Average Action (EAA) $\Gamma_k$. For our purposes, however, in Section 3 we are led to consider the generalised form of the flow of the EAA which incorporates frame transformations along the RG flow [8]. It is this equation that allows us to implement essential schemes. Moreover, we derive the dimensionless form of the generalised flow equation, where it becomes clear that the cutoff scale $k$ is itself an inessential coupling. We notice that the RG equations we use can be seen as the counterpart of the generalised flow equations for the Wilsonian effective action first written down by F. Wegner [15].

In order to make contact with the previous versions of the exact RG, in Section 4 we reduce our general equations to the *standard scheme* where only a single inessential coupling, namely the wave function renormalisation, is specified.

Having presented the frame covariant formulation of the exact RG, in Section 5 we introduce the minimal essential scheme. In this scheme, all the inessential couplings are set to zero at every scale along the RG flow. Several comments are in order. Having a scheme of this type at hand provides practical advantages as well as a clearer physical picture of renormalisation. On the practical side, a major improvement of the minimal essential scheme as compared to the standard one is the fact that the form of the propagator maintains a simple form along the RG flow. This ensures that the propagating degrees of freedom are just those of the corresponding free theory. Conceptually, our scheme may also lead to a better understanding of the equivalence of quantum field theories [16–18] and the universality of statistical physics models at criticality, building on the insights of previous works [1, 2, 15, 19–23]. Moreover, we further develop and take advantage of the analogy between frame transformations and gauge transformations [20]. Although, for the sake of simplicity, we will treat a single scalar field $\phi$, the generalisation to theories with other field content is obvious. As such, the scheme which we develop can be exploited in a wide range of areas of theoretical physics where the exact RG is a useful calculation tool.

F. Wegner proved [15] that, at a fixed point of the RG, critical exponents associated with redundant operators are entirely scheme-dependent. Section 6 is then devoted to the discussion of the fixed-point equations and how the corresponding critical exponents can be obtained, contrasting the differences between the standard and (minimal) essential schemes. In particular, we pay attention to the identification of the anomalous dimension, whose computation presents the most substantial differences with respect to the standard case. One of the most prominent results in this Section regards the fact that at a fixed point, redundant perturbations are automatically discarded. This makes essential schemes a preferred tool to access only the necessary, essential physical content.

Moving towards actual implementations of essential schemes, it is important to realise that, a priori, the EAA may contain all possible terms compatible with the symmetries of the model under consideration. However, any concrete application of the exact RG relies on approximation schemes that reduce the EAA to a manageable subset of all terms. The celebrated *derivative expansion* [24,25] consists of approximating $\Gamma_k[\phi]$ by its Taylor expansion in gradients of $\phi$. In this manner, in order to obtain approximate beta functions with a finite amount of effort, one typically has to truncate the derivative expansion to

168 a given finite order $\partial^s$. At each order $s = 0, 2, 4, \ldots$ one is able to compute physical
169 quantities, providing estimates which show convergence as $s$ is increased. To date, this
170 program has been carried out in the standard scheme up to order $s = 6$ for the 3D Ising
171 model [26], where furthermore it has been argued that the derivative expansion can have
172 a finite radius of convergence. While at order $s = 0$ the EAA is projected onto the space of
173 effective potentials $V_k(\phi)$ [27, 28], at higher orders, one obtains coupled flow equations for
174 an increasing number of independent functions of the field [25, 26, 29–31]. Consequently,
175 as the order increases, this program rapidly grows in complexity. The minimal essential
176 scheme reduces this complexity order by order in the derivative expansion. In addition,
177 while there can be spurious effects due to approximations, those arising from inessential
178 couplings will not be present.

179     To demonstrate the scheme's utility, in Section 7 we derive the explicit form of the
180 flow equation at order $s = 2$ of the derivative expansion and in Section 8 we apply it to
181 the study of the critical point of the 3D Ising model. In particular, we shall identify the
182 Wilson-Fisher fixed point as a globally-defined scaling solution to the exact RG equations
183 and calculate the values of the universal critical exponents $\nu$, $\omega$ and $\eta$. These results are
184 obtained by solving the flow equations both functionally and with a polynomial truncation.
185 The numerical estimates we obtained for the critical exponents are found to be in good
186 agreement w.r.t. the computations performed at order $\partial^2$ in the standard scheme [30, 32–
187 34]. The simplifications exemplified by this application of the minimal essential scheme
188 at order $s = 2$ of the derivative expansion are expected at all higher orders. This is
189 demonstrated in Section 9 by providing a recipe on how to implement the minimal essential
190 scheme order by order.

191     We devote Sections 10 to a general discussion: here we advocate the possibility of em-
192 ploying non-minimal essential schemes in optimisation problems by applying extended
193 principle of minimal sensitivity (PMS) studies [35]. After taking the opportunity to
194 make general considerations about redundant operators and the generalisability of essen-
195 tial schemes, we then discuss the implications entailed for asymptotic safety in quantum
196 gravity and for the frame equivalence problem in Cosmology. Conclusions are finally pro-
197 vided in Section 11. Appendix A contains a detailed derivation of the frame covariant
198 exact renormalisation group equation for the EAA. In Appendix B we show some iden-
199 tities related to the generator of dilatations, which are important to express the exact
200 renormalisation flow equations in dimensionless variables. In Appendix C we comment
201 on the connection between the renormalisation conditions and inessential couplings for
202 free theories including the high temperature fixed point and higher-derivative theories.
203 Finally, in Appendix D we explicitly calculate the general flow equation at second order in
204 derivative expansion in two different ways, i.e. in momentum space and in position space.

## 2  Frame transformations in quantum field theory

### 2.1  Classical frame transformations

207 The classical dynamics of a field theory is encoded in an action $S_\chi[\chi]$. This can be
208 considered as a scalar function on the configuration space $\mathcal{M}$ viewed a manifold, where
209 the points are field configurations $\chi : \mathbb{R}^d \to \mathbb{R}$. In this respect, the values of the dynamical
210 field variable $\chi(x)$ can be considered as a preferred coordinate system for which the action
211 takes a particular form. What distinguishes the variable $\chi$ as "the field" is that, typically,
212 it assumes a straightforward physical significance being an easily accessible observable
213 experimentally. From a geometrical point of view, this is equivalent to defining a particular

local set of *frames* on $\mathcal{M}$. The classical dynamics is then defined by the principle that the action is stationary, namely

$$\frac{\delta S_\chi}{\delta \chi(x)} = 0 \, . \tag{1}$$

This provides the equations of motion for the field variable $\chi$. However, it could be the case that the equations of motion are relatively difficult to solve when written in terms of $\chi$ and can be simplified by re-expressing the action in terms of different variables $\phi = \phi[\chi]$. Provided the map $\phi[\chi]$ is invertible, such that the inverse map $\chi = \chi[\phi]$ exists, this amounts to choosing a different frame. If this is the case, we can solve the equations of motion for a new action $S_\phi[\phi]$, which is related to the action in the original frame by

$$S_\chi[\chi] = S_\phi[\phi[\chi]] \, . \tag{2}$$

The solutions to the two equations of motion are then in a one-to-one correspondence since invertibility ensures that the Jacobian between the two frames is non-singular. To see this correspondence, we observe that (1) can be written as[1]

$$\int_{x_1} \frac{\delta \phi(x_1)}{\delta \chi(x)} \frac{\delta S_\phi[\phi]}{\delta \phi(x_1)} = 0 \, , \tag{3}$$

and, as such, the non-singular nature of the Jacobian implies that

$$\frac{\delta S_\phi[\phi]}{\delta \phi(x)} = 0 \, . \tag{4}$$

To calculate observables, we should evaluate them on the dynamical shell consisting of points on $\mathcal{M}$ where (1) is satisfied. However, one should bear in mind that observables transform as scalars on $\mathcal{M}$, and therefore, they must transform accordingly.

In general the map $\phi[\chi]$ can be non-linear in the field $\chi$. The imposition that $\phi[\chi]$ is invertible in the vicinity of a constant field configuration also restricts the map to be *quasi-local*. Specifically, quasi-local means that if we expand $\phi[\chi]$ in derivatives of the field, the expansion is analytic and thus we can write

$$\phi(x) \sim \sum_{s=0}^{\infty} L_s(\chi(x), \partial_\mu \chi(x), \dots) \, , \tag{5}$$

where $L_s = O(\partial^s)$ are local functions of the field and its derivatives at $x$, involving $s$ derivatives. If the series terminates at a finite order then we have strict locality.

As an example of a frame transformation, let us consider a generic action involving up to two derivatives of the field

$$S_\chi[\chi] = \int_x \left[ \frac{z_\chi(\chi)}{2} (\partial_\mu \chi)(\partial_\mu \chi) + V_\chi(\chi) \right] \, , \tag{6}$$

this can be re-expressed in the *canonical frame* where it depends only on a potential $V_\phi(\phi) = V_\chi(\chi(\phi))$, assuming therefore the simpler form

$$S_\phi[\phi] = \int_x \left[ \frac{1}{2} (\partial_\mu \phi)(\partial_\mu \phi) + V_\phi(\phi) \right] \, . \tag{7}$$

This is achieved by the following transformation

$$\chi \to \chi(\phi) \, , \quad \frac{\partial \chi(\phi)}{\partial \phi} = \frac{1}{\sqrt{z_\chi(\chi(\phi))}} \, , \tag{8}$$

---

[1]Hereafter we use the shorthand notation $\int_x := \int \mathrm{d}^d x$.

which is the inverse of the transformation

$$\phi \to \phi(\chi)\,, \quad \frac{\partial\phi(\chi)}{\partial\chi} = \sqrt{z_\chi(\chi)}\,. \tag{9}$$

Thus, provided $z_\chi(\chi)$ is non-singular, we can transform to the canonical frame where solutions to the equations of motion will be in a one-to-one correspondence.

More generally, actions in two different frames will transform as scalars on $\mathcal{M}$, where a change of frame is understood as a diffeomorphism from $\mathcal{M}$ to itself. Under an infinitesimal frame transformation $\phi \to \phi + \xi[\phi]$, the action transforms as

$$S[\phi] \to S[\phi] + \xi[\phi] \cdot \frac{\delta}{\delta\phi} S[\phi]\,, \tag{10}$$

where, hereafter, we adopt the condensed notation for which a dot implies an integral over $x$ such that $X \cdot Y := \int_x X(x)Y(x)$. For definiteness, we consider the field to have a single component, however, the generalisation to a multi-component field $\phi^A(x)$ is straightforward since the dot would then also imply a sum over the components $X \cdot Y := \sum_A \int_x X_A(x)Y_A(x)$.

The transformation (10) is an infinitesimal *classical* frame transformation. It is clear that, with a bit of work, classical field theory can be formulated in a covariant language allowing one the freedom to easily pick different frames to calculate observables. This freedom is analogous to the freedom to pick a particular gauge condition in general relativity, which amounts to picking a set of local frames on spacetime. In the rest of this Section, we lift the discussion on frame transformations in order to develop a frame covariant formulation of quantum field theory.

## 2.2 The principle of frame invariance in QFT

In quantum field theory (QFT), all physical information is stored in correlation functions. In the path-integral formalism, these are functionals $\hat{\mathcal{O}}[\hat{\chi}]$ of the quantum field $\hat{\chi}$ averaged over all possible field configurations (quantum fluctuations), in which each configuration is weighted with $e^{-S}$. Therefore, the most general objects which we wish to compute are expectation values of *observables* $\hat{\mathcal{O}}$ given by

$$\langle\hat{\mathcal{O}}\rangle := \mathcal{N} \int (\mathrm{d}\hat{\chi})\, \hat{\mathcal{O}}_{\hat{\chi}}[\hat{\chi}]\, e^{-S_{\hat{\chi}}[\hat{\chi}]}\,, \tag{11}$$

where $\mathcal{N}^{-1} = \int (\mathrm{d}\hat{\chi})\, e^{-S_{\hat{\chi}}[\hat{\chi}]}$ and $\hat{\mathcal{O}}_{\hat{\chi}}[\hat{\chi}] = \hat{\mathcal{O}}$ is an observable expressed as functional of the fields $\hat{\chi}$, which in general can be an $n$-point function. For example we could be interested in an 2-point function of the field in which case

$$\hat{\mathcal{O}}_{\hat{\chi}}[\hat{\chi}] = \hat{\chi}(x_1)\hat{\chi}(x_2)\,, \tag{12}$$

but we could also be interested in products of composite operators at different points in space.

The exact definition of the path integral measure depends on the regularisation. For the class of regulators which we employ, it is defined by

$$\int (\mathrm{d}\hat{\chi})\, e^{-\frac{1}{2}\hat{\chi}\cdot M_\Lambda \cdot \hat{\chi}} = 1\,, \tag{13}$$

where $\Lambda$ is the ultraviolet cutoff which we will formally take to infinity or to some scale much greater than all relevant physical scales. The two-point function $M_\Lambda(x_1, x_2)$ can be

understood as a metric on $\mathcal{M}$ which is independent of the field $\hat{\chi}$ and should diverge in the continuum limit, namely

$$\lim_{\Lambda \to \infty} M_\Lambda \to \infty \,. \tag{14}$$

In the simplest case, $M_\Lambda(x_1, x_2) = \alpha \Lambda^2 \delta(x_1, x_2)$, where $\alpha$ is a positive constant.

In practice, the computation of correlation functions is facilitated by the introduction of suitable generating functionals. For example, the generating functional $\mathcal{W}_{\hat{\chi}}[J]$ of the (connected) correlation functions for the field $\hat{\chi}$ is given by

$$\mathcal{N} \, \mathrm{e}^{\mathcal{W}_{\hat{\chi}}[J]} := \langle \mathrm{e}^{J \cdot \hat{\chi}} \rangle = \mathcal{N} \int (\mathrm{d}\hat{\chi}) \, \mathrm{e}^{J \cdot \hat{\chi}} \mathrm{e}^{-S_{\hat{\chi}}[\hat{\chi}]} \,, \tag{15}$$

where $J \cdot \hat{\chi}$ is a source term for the field $\hat{\chi}$. Here we are interested in the generalisation of (15) where the source $J$ couples instead to a composite operator $\hat{\phi} = \hat{\phi}[\hat{\chi}]$, such that we generate the correlation functions of $\hat{\phi}$ rather than those of $\hat{\chi}$. To ensure that these correlation functions contain the same physical information, we take $\hat{\phi} = \hat{\phi}[\hat{\chi}]$ to define a diffeomorphism from $\mathcal{M}$ to itself, or phrased differently, a frame transformation from the original $\hat{\chi}$-frame to a new $\hat{\phi}$-frame. Therefore, we are led to consider a family of generating functionals

$$\mathcal{N} \, \mathrm{e}^{\mathcal{W}_{\hat{\phi}}[J]} := \langle \mathrm{e}^{J \cdot \hat{\phi}} \rangle = \mathcal{N} \int (\mathrm{d}\hat{\chi}) \, \mathrm{e}^{J \cdot \hat{\phi}[\hat{\chi}]} \mathrm{e}^{-S_{\hat{\chi}}[\hat{\chi}]} \,, \tag{16}$$

for the composite operator $\hat{\phi}[\hat{\chi}]$, which from now on we call the *parameterised field*. In geometrical terms, (16) makes sense if we understand $\hat{\phi}(x)$ as a set of scalars on $\mathcal{M}$ labelled by the points in real space $x$. If we were to introduce purely abstract coordinates on $\mathcal{M}$, then the gradient of $\hat{\phi}(x)$ is a coframe field while the inverse the coframe field is a frame field.

In presence of the source, expectation values are given by

$$\langle \hat{\mathcal{O}} \rangle_J = \mathrm{e}^{-\mathcal{W}_{\hat{\phi}}[J]} \langle \mathrm{e}^{J \cdot \hat{\phi}} \hat{\mathcal{O}} \rangle \,, \tag{17}$$

and they reduce to (11) by taking $J = 0$. In practice, given (16), source-dependent expectation values can be computed as

$$\langle \hat{\mathcal{O}} \rangle_J = \mathrm{e}^{-\mathcal{W}_{\hat{\phi}}[J]} \hat{\mathcal{O}} \Big[ \hat{\chi} \Big[ \frac{\delta}{\delta J} \Big] \Big] \mathrm{e}^{\mathcal{W}_{\hat{\phi}}[J]} \,, \tag{18}$$

where $\hat{\chi}[\hat{\phi}]$ is the inverse diffeomorphism of $\hat{\phi}$. Since the observables $\hat{\mathcal{O}}$ are scalars on $\mathcal{M}$, such that

$$\hat{\mathcal{O}} = \hat{\mathcal{O}}_{\hat{\chi}}[\hat{\chi}] = \hat{\mathcal{O}}_{\hat{\phi}}[\hat{\phi}] \,, \tag{19}$$

we can thus equivalently write (18) as

$$\langle \hat{\mathcal{O}} \rangle_J = \mathrm{e}^{-\mathcal{W}_{\hat{\phi}}[J]} \hat{\mathcal{O}}_{\hat{\phi}} \Big[ \frac{\delta}{\delta J} \Big] \mathrm{e}^{\mathcal{W}_{\hat{\phi}}[J]} \,. \tag{20}$$

The source $J$ could be viewed as a physical external field that couples linearly to $\hat{\phi}$. In this interpretation, however, we would be considering a model where $S_{\hat{\chi}}[\hat{\chi}]$ is replaced by $S_{\hat{\chi}}[\hat{\chi}] - J \cdot \hat{\phi}[\hat{\chi}]$, resulting in a physical dependence on the choice of frame. In this paper, instead, we will adopt the *principle of frame invariance*, meaning that we will work within a frame covariant (or other words reparameterisation, or field-redefinition covariant) formalism where physical quantities are independent of the choice of frame. Consequently, in this formalism all physical couplings, possibly including a coupling $h \cdot \hat{\chi}$ to an external field $h$, should be part of the action $S_{\hat{\chi}}$, and the source $J$ shall be viewed merely as a device to compute correlation functions such that, after differentiating $\mathcal{W}_{\hat{\phi}}[J]$,

we are ultimately interested in taking $J = 0$. Physical quantities are therefore obtained by the frame covariant expression[2]

$$\langle \hat{\mathcal{O}} \rangle = e^{-\mathcal{W}[J]} \hat{\mathcal{O}} \left[ \frac{\delta}{\delta J} \right] e^{\mathcal{W}[J]} \bigg|_{J=0} , \tag{21}$$

with the final result being a frame invariant quantity. For example the 2-point functions is obtained by

$$\langle \hat{\chi}(x_1) \hat{\chi}(x_2) \rangle = e^{-\mathcal{W}[J]} \hat{\chi} \left[ \frac{\delta}{\delta J(x_1)} \right] \hat{\chi} \left[ \frac{\delta}{\delta J(x_2)} \right] e^{\mathcal{W}[J]} \bigg|_{J=0} , \tag{22}$$

The advantage of working with a frame covariant setup is that the complexity of computing certain physical quantities may be reduced by the choice of a specific frame. For many quantities such as the correlation functions of the physical field $\hat{\chi}$ e.g. (22), the specific choice of the frame may simply be $\hat{\phi} = \hat{\chi}$. However, for universal quantities computed in the vicinity of a continuous phase transition in statistical physics, or quantities which are computed at vanishing external field, such as S-matrix elements in particle physics, it may be that the specific choice of $\hat{\phi}$ is non-trivial. What is important is that in principle we can compute any observable in any frame. Then in practice we can exploit the frame where computations become most manageable.

## 2.3 Change of integration variables

In addition to the freedom of fixing a frame by choosing a particular $\hat{\phi}[\hat{\chi}]$ which couples to the source, we are also at liberty to make a change of integration variables in the corresponding functional integral (16). Under this change of variables, $\hat{\phi}[\hat{\chi}]$ transforms as a set of scalars on $\mathcal{M}$ and $\mathcal{W}_{\hat{\phi}}[J]$ is hence invariant. Of course, we are at liberty to make $\hat{\phi}$ the integration variable and therefore we can equivalently write

$$e^{\mathcal{W}_{\hat{\phi}}[J]} = \int (d\hat{\phi}) \, e^{-S_{\hat{\phi}}[\hat{\phi}]} \, e^{J \cdot \hat{\phi}} , \tag{23}$$

where

$$e^{-S_{\hat{\phi}}[\hat{\phi}]} = e^{-S_{\hat{\chi}}[\hat{\chi}[\hat{\phi}]]} \det \frac{\delta \hat{\chi}[\hat{\phi}]}{\delta \hat{\phi}} \tag{24}$$

has transformed as a density. However, since these transformations leave $\mathcal{W}[J]$ invariant, it is entirely immaterial whether we perform this transformation (or any other change of integration variables) or not. Furthermore, the expectation value of an observable (i.e. what we mean by $\langle \dots \rangle$) can also be defined in a covariant way as

$$\langle \hat{\mathcal{O}} \rangle := \mathcal{N} \int (d\hat{\phi}) \, \hat{\mathcal{O}}_{\hat{\phi}}[\hat{\phi}] \, e^{-S_{\hat{\phi}}[\hat{\phi}]} , \tag{25}$$

which is equivalent to the previous definition (11). In this paper, by a frame transformation, we always refer to a change in the field which couples to the source, rather than a change of integration variables.

---

[2]From now on we can suppress the $\hat{\phi}$ subscripts from $\mathcal{W}[J] \equiv \mathcal{W}_{\hat{\phi}}[J]$, $\hat{\mathcal{O}}[\hat{\phi}] \equiv \hat{\mathcal{O}}_{\hat{\phi}}[\hat{\phi}]$ etc. whenever we are discussing a generic frame and no confusion can arise.

### 2.4 Effective actions

Given $\mathcal{W}[J]$, other generating functionals, related to $\mathcal{W}[J]$ by transformations and/or the addition of further sources, can be considered. For example, the one-particle irreducible (1PI) effective action $\Gamma[\phi]$ is obtained by the Legendre transform

$$\Gamma_{\hat{\phi}}[\phi] = -\mathcal{W}_{\hat{\phi}}[J] + \phi \cdot J \,, \tag{26}$$

where $\phi = \langle \hat{\phi}[\hat{\chi}] \rangle_J$ is the mean parameterised field. Equivalently, $\Gamma[\phi]$ can be defined by the solution to the integro-differential equation

$$\mathcal{N} \, \mathrm{e}^{-\Gamma[\phi]} = \langle \mathrm{e}^{(\hat{\phi}-\phi)\cdot\frac{\delta}{\delta\phi}\Gamma[\phi]} \rangle \,, \tag{27}$$

with $\phi$-dependent expectation values given by

$$\langle \hat{\mathcal{O}}[\hat{\chi}] \rangle_\phi = \mathrm{e}^{\Gamma[\phi]} \langle \mathrm{e}^{(\hat{\phi}-\phi)\cdot\frac{\delta}{\delta\phi}\Gamma} \hat{\mathcal{O}}[\hat{\chi}] \rangle \,. \tag{28}$$

For our purposes, we will be interested in a particular class of generating functionals that generalise the 1PI effective action in the presence of an additional source $K(x_1, x_2)$ for two-point functions. In the next Section we will identify $K(x_1, x_2)$ with a cutoff function, but for now, we view it simply as an additional source independent of $\phi$. Its inclusion leads to a modified effective action

$$\mathcal{N} \, \mathrm{e}^{-\Gamma[\phi,K]} = \langle \mathrm{e}^{(\hat{\phi}-\phi)\cdot\frac{\delta}{\delta\phi}\Gamma[\phi,K]-\frac{1}{2}(\hat{\phi}-\phi)\cdot K\cdot(\hat{\phi}-\phi)} \rangle \,. \tag{29}$$

so that $K$- and $\phi$-dependent expectation values can be defined by

$$\langle \hat{\mathcal{O}} \rangle_{\phi,K} = \mathrm{e}^{\Gamma[\phi,K]} \langle \mathrm{e}^{(\hat{\phi}-\phi)\cdot\frac{\delta}{\delta\phi}\Gamma[\phi,K]-\frac{1}{2}(\hat{\phi}-\phi)\cdot K\cdot(\hat{\phi}-\phi)} \hat{\mathcal{O}} \rangle \,. \tag{30}$$

We will also denote the expectation value of an operator $\hat{\mathcal{O}}$ by dropping the hat, such that

$$\mathcal{O}[\phi,K] \equiv \langle \hat{\mathcal{O}} \rangle_{\phi,K} \,. \tag{31}$$

### 2.5 Functional identities

An infinite set of identities can be derived systematically by taking successive derivatives of (29) and (30) with respect to $\phi$ and $K$ and using the identities obtained from lower derivatives. Here we will obtain those identities which we will make explicit use of in the rest of the paper. First, taking one derivative of (29) with respect to $\phi$ one finds that

$$(K + \Gamma^{(2)}[\phi,K]) \cdot (\phi - \langle \hat{\phi} \rangle_{\phi,K}) = 0 \,, \tag{32}$$

where $\Gamma^{(2)}[\phi,K]$ denotes the second functional derivative of $\Gamma[\phi,K]$ with respect to $\phi$. Thus, assuming the invertibility of $K + \Gamma^{(2)}[\phi,K]$, one has that $\phi$ is again the mean parameterised field

$$\phi = \langle \hat{\phi} \rangle_{\phi,K} \,. \tag{33}$$

Taking a further derivative of (33) with respect to $\phi$ one finds that the two-point function is given by

$$\begin{aligned}
\mathcal{G}_{x_1,x_2}[\phi,K] &:= \langle (\hat{\phi}(x_1) - \phi(x_1))(\hat{\phi}(x_2) - \phi(x_2)) \rangle_{\phi,K} \\
&= \frac{1}{\Gamma^{(2)}[\phi,K] + K}(x_1, x_2) \,.
\end{aligned} \tag{34}$$

Then, varying (29) with respect to $K$ at fixed $\phi$ we obtain the functional identity [13, 14]

$$\delta\Gamma[\phi, K]\big|_\phi = \frac{1}{2}\mathrm{Tr}\,\mathcal{G}[\phi, K] \cdot \delta K\,, \tag{35}$$

where $\mathrm{Tr}$ stands for the trace of a two-point function $\mathrm{Tr}\,X := \int_x X(x, x)$. Taking a functional derivative of (30) with respect to $\phi$ and using the previously derived identities we obtain

$$\langle(\hat{\phi} - \phi)\hat{\mathcal{O}}\rangle_{\phi, K} = \mathcal{G}[\phi, K] \cdot \frac{\delta}{\delta\phi}\mathcal{O}[\phi, K]\,. \tag{36}$$

There are two special configurations of the source $K(x_1, x_2)$. First, if we take $K = 0$ then $\Gamma[\phi, 0] = \Gamma[\phi]$ is the 1PI effective action. If additionally $\Gamma[\phi]$ is evaluated at its stationary point $\phi_{\min}$ the expectation values (30) reduce to the frame invariants (11). Secondly, if we take $K(x_1, x_2) = M_\Lambda(x_1, x_2)$, where $M_\Lambda$ is the metric that defines the measure (13), then the two-point source term produces a delta function in the path integral as the continuum limit (14) is taken, and we have

$$\lim_{\Lambda \to \infty} \Gamma[\phi, M_\Lambda] = S[\phi]\,, \tag{37}$$

where $S[\phi] = S_{\hat{\phi}}[\phi]$ is given by (24). Furthermore, the expectation values are given by the mean-field expression

$$\lim_{\Lambda \to \infty} \langle\hat{\mathcal{O}}\rangle_{\phi, M_\Lambda} = \hat{\mathcal{O}}[\phi]\,. \tag{38}$$

It is these two limits that make $\Gamma[\phi, K]$ a useful generating functional for the exact RG since one can realise Wilson's concept of an incomplete integration by allowing $K$ to interpolate between the limits.

## 2.6 Inessential couplings and active frame transformations

Although in a particular frame the microscopic action may assume a relatively simple form, e.g. $S_{\hat{\chi}}[\hat{\chi}] = \int_x \left[\frac{1}{2}(\partial_\mu\hat{\chi})(\partial_\mu\hat{\chi}) + \frac{1}{2}m^2\hat{\chi}^2 + \frac{1}{4!}\lambda\hat{\chi}^4\right]$, the generating functionals will typically be very complicated. As a consequence of this, expanding the generating functionals in a typical operator basis, we will find an infinite set of non-vanishing coupling constants $g_i$. These couplings can be viewed coordinates on theory space. Different choices of the operator basis in terms of which we expand the generating functionals, therefore, correspond to different coordinate systems on theory space (for a discussion on the geometry of theory space see [36]). In a frame covariant formalism, we are free to make frame transformations without affecting physical observables even though the form of the generating functionals will change. Consequently, any change in the coupling constants[3] $g_i \to g_i + \delta g_i$ which is equivalent to a frame transformation gives a theory that is physically equivalent to the original theory. Put differently, there are directions in theory space along which all physical quantities remain unchanged. These directions form 'sub-manifolds of constant physics' in theory space. Locally in theory space, we can therefore work in a coordinate system $\{g_i\} = \{\lambda_a, \zeta_\alpha\}$ adapted to these sub-manifolds where $\lambda_a$ are the essential couplings which will appear in expressions for the physical observables (11). The remaining couplings $\zeta_\alpha$ are therefore the inessential couplings. It follows that changing the values of the inessential couplings $\zeta \to \zeta + \delta\zeta$ is equivalent to the change induced by a local frame transformation

$$\hat{\phi}[\hat{\chi}] \to \hat{\phi}[\hat{\chi}] - \hat{\xi}[\hat{\chi}] + O(\hat{\xi}^2)\,, \tag{39}$$

---

[3]Here we are using $\delta$ to denote a variation with respect to the couplings keeping field variables fixed.

where $\hat{\xi}[\hat{\chi}] = \hat{\Phi}[\hat{\chi}]\,\zeta\delta\zeta$. For the generating functionals $\mathcal{W}[J]$, $\Gamma[\phi]$ and $\Gamma[\phi, K]$ one finds that they transform respectively as

$$\mathcal{W}[J] \to \mathcal{W}[J] - J \cdot \xi[J] + O(\xi^2)\,, \tag{40}$$

$$\Gamma[\phi] \to \Gamma[\phi] + \xi[\phi] \cdot \frac{\delta}{\delta\phi}\Gamma[\phi] + O(\xi^2)\,, \tag{41}$$

$$\Gamma[\phi, K] \to \Gamma[\phi, K] + \xi[\phi, K] \cdot \frac{\delta}{\delta\phi}\Gamma[\phi, K] - \operatorname{Tr}\mathcal{G}[\phi, K] \cdot \frac{\delta}{\delta\phi}\xi[\phi, K] \cdot K + O(\xi^2)\,, \tag{42}$$

where $\xi[J]$, $\xi[\phi]$ and $\xi[\phi, K]$ are expectation values

$$\xi[J] = \langle\hat{\xi}[\hat{\chi}]\rangle_J\,, \tag{43}$$

$$\xi[\phi] = \langle\hat{\xi}[\hat{\chi}]\rangle_\phi\,, \tag{44}$$

$$\xi[\phi, K] = \langle\hat{\xi}[\hat{\chi}]\rangle_{\phi, K}\,. \tag{45}$$

In (42) the form of the term involving the trace comes from using the identity (36) with $\hat{\mathcal{O}} = \hat{\xi}$.

In the case of the 1PI effective action $\Gamma[\phi]$ we note that (41) has the same form as the classical frame transformation (10). This means that a derivative of $\Gamma[\phi]$ with respect to an inessential coupling gives

$$\zeta\frac{\partial}{\partial\zeta}\Gamma[\phi] = \Phi[\phi] \cdot \frac{\delta}{\delta\phi}\Gamma[\phi]\,, \tag{46}$$

for some $\Phi[\phi]$. We see explicitly that the frame transformation is proportional to the equation of motion as in the classical case. This is the origin of the statement that one can use the equations of motion to calculate the running of essential couplings [2]. However, in what follows we will work with the EAA, which has the form of $\Gamma[\phi, K]$ where $K$ is chosen to be a cutoff function. In this case, therefore, we have that

$$\zeta\frac{\partial}{\partial\zeta}\Gamma[\phi, K] = \Phi[\phi, K] \cdot \frac{\delta}{\delta\phi}\Gamma[\phi, K] - \operatorname{Tr}\mathcal{G}[\phi, K] \cdot \frac{\delta}{\delta\phi}\Phi[\phi, K] \cdot K\,. \tag{47}$$

We see that this transformation includes a loop term in addition to the tree-level term which vanishes on the equation of motion. The operator on the r.h.s. of (47) is the *re-dundant operator* conjugate to the inessential coupling $\zeta$. Every inessential coupling is therefore conjugate to a redundant operator which is in turn determined by some (quasi-)local field $\Phi(x)$ which characterises the frame transformation. From a geometrical point of view, a derivative with respect to an inessential coupling can be understood as an "averaged" Lie derivative. While $\Gamma[\phi]$ is in this sense a scalar, the averaged Lie derivative of $\Gamma[\phi, K]$ is non-linear due to the presence of $K$. From this point of view, (47) can be understood as an *active frame transformation* (or active reparameterisation), where the functional form of $\Gamma[\phi, K]$ is modified leaving $\phi$ and $K$ fixed. An active frame transformation is therefore equivalent to a change in the values of the inessential couplings keeping the essential couplings fixed. Different frames are therefore fully characterised by specifying values of the inessential couplings. The analogy with gauge fixing in general relativity is then clear: the frame transformations are analogous to gauge transformations while conditions that specify the inessential couplings are analogous to gauge fixing conditions.

## 2.7 Passive frame transformations

Instead of active frame transformations, we can consider *passive frame transformations*, namely those which are characterised by simply expressing $\Gamma[\phi, K]$ in terms of different

variables. These will not be simply related to active frame transformations since, for a non-linear function $\Phi[\phi] \neq \langle \Phi[\hat{\phi}] \rangle$. However, if we consider a linear frame transformation of the form

$$\hat{\phi}'' = c \cdot \hat{\phi}', \tag{48}$$

where $c$ is a field independent two-point function, one has that $\phi'' = c \cdot \phi'$. From this property, we have the simple identity

$$\Gamma_{\hat{\phi}'}[\phi', c^T \cdot K \cdot c] = \Gamma_{\hat{\phi}''}[c \cdot \phi', K], \tag{49}$$

where $c^T$ is the transpose of $c$. These linear passive frame transformations will help us to make contact with more standard derivations of the exact RG equation and clarify the transition from dimensionless to dimensionful variables. More generally, they expose the fact that a linear transformation of $K$ and $\phi$ which keeps $\phi \cdot K \cdot \phi$ invariant is equivalent to a frame transformation.

# 3  Frame covariant flow equation

We will now write down RG flow equations for a frame covariant EAA. These will take a generalised form which will allow us to make arbitrary frame transformations along an RG trajectory. The equations can be written both in dimensionful variables, where the cutoff scale $k$ is made explicit or in dimensionless variables, where we work in units of $k$ and hence all the quantities including the coordinates $y := kx$ are dimensionless. The dimensionful version (56), along with more general flow equations which incorporate field redefinitions along the flow, has been derived previously in [8].

## 3.1  Dimensionful covariant flow

In dimensionful variables, the frame covariant effective average action is obtained by introducing a cutoff scale $k$ in two independent manners. Firstly, we identify $K = \mathcal{R}_k$ with an additive IR cut off $\mathcal{R}_k$ which suppresses fluctuations below momentum scales $p^2 \simeq k^2$ and vanishes in the ultraviolet (UV) for momenta $p^2 \gg k^2$. In position space the regulator is a function of the Bochner-Laplacian $\Delta = -\partial_\mu \partial_\mu$ such that[4]

$$\begin{aligned} \mathcal{R}_k(x_1, x_2) &= k^2 \, R(\Delta/k^2) \delta(x_1, x_2) \\ &= k^2 \int_p R(p^2/k^2) \, \mathrm{e}^{\mathrm{i}p_\mu(x_1^\mu - x_2^\mu)}, \end{aligned} \tag{50}$$

where $R(p^2/k^2)$ is the dimensionless cutoff function which vanishes in the limit $p^2/k^2 \to \infty$, while for $p^2/k^2 \to 0$ it has a non-zero limit $R(0) > 0$, ensuring the suppression of IR modes. Secondly, one allows the parameterised field $\hat{\phi}$ itself to depend on $k$. This leads to the following frame covariant effective average action

$$\mathcal{N} \mathrm{e}^{-\Gamma_k[\phi]} := \langle \mathrm{e}^{(\hat{\phi}_k - \phi) \cdot \frac{\delta}{\delta \phi} \Gamma_k[\phi] - \frac{1}{2}(\hat{\phi}_k - \phi) \cdot \mathcal{R}_k \cdot (\hat{\phi}_k - \phi)} \rangle, \tag{51}$$

which is the effective action (29), where the source for the two-point functions $K$ is now specified to be given by the cutoff function $\mathcal{R}_k$ and where $\hat{\phi} = \hat{\phi}_k[\hat{\chi}]$ is the $k$-dependent parameterised field. Therefore an equivalent definition is

$$\Gamma_k[\phi] = \Gamma_{\hat{\phi}_k}[\phi, \mathcal{R}_k], \tag{52}$$

---

[4]Where we adopt the following notation $\int_p := \int \frac{\mathrm{d}^d p}{(2\pi)^d}$.

where the $k$ dependence of $\Gamma_k[\phi]$ comes from both the $k$ dependence of the regulator $\mathcal{R}_k$ and the parameterised field $\hat{\phi}_k$. We can then define $k$- and $\phi$-dependent expectation in the usual manner, namely

$$\langle \hat{\mathcal{O}} \rangle_{\phi,k} = e^{\Gamma_k[\phi]} \langle e^{(\hat{\phi}_k - \phi) \cdot \frac{\delta}{\delta\phi} \Gamma_k[\phi] - \frac{1}{2}(\hat{\phi}_k - \phi) \cdot \mathcal{R}_k \cdot (\hat{\phi}_k - \phi)} \hat{\mathcal{O}} \rangle , \tag{53}$$

such that in this case the general identity (33) implies

$$\phi = \langle \hat{\phi}_k \rangle_{\phi,k} . \tag{54}$$

Here we anticipate that letting the parameterised field $\hat{\phi}_k$ to be itself $k$-dependent, allows for the possibility of eliminating all the inessential coupling constants from the set of independent running couplings. This, in a nutshell, will be what we define later as an *essential scheme*. In this respect, we recognise that the redundant operators assume the following form

$$\zeta \frac{\partial}{\partial\zeta} \Gamma_k[\phi] = \Phi_k[\phi] \cdot \frac{\delta}{\delta\phi} \Gamma_k[\phi] - \mathrm{Tr}\, \mathcal{G}_k[\phi] \cdot \frac{\delta}{\delta\phi} \Phi_k[\phi] \cdot \mathcal{R}_k , \tag{55}$$

where $\mathcal{G}_k[\phi] = (\Gamma_k^{(2)}[\phi] + \mathcal{R}_k)^{-1}$ is the IR regularised propagator. The exact RG flow equation obeyed by the frame covariant EAA (51) is then given by

$$\boxed{\left( \partial_t + \Psi_k[\phi] \cdot \frac{\delta}{\delta\phi} \right) \Gamma_k[\phi] = \frac{1}{2} \mathrm{Tr}\, \mathcal{G}_k[\phi] \left( \partial_t + 2 \cdot \frac{\delta}{\delta\phi} \Psi_k[\phi] \right) \cdot \mathcal{R}_k} , \tag{56}$$

where $t \coloneqq \log(k/k_0)$, with $k_0$ some physical reference scale, and

$$\Psi_k[\phi] \coloneqq \langle \partial_t \hat{\phi}_k[\hat{\chi}] \rangle_{\phi,k} \tag{57}$$

is the *RG kernel* which can be a general quasi-local functional of the field $\phi$. The flow equation (56) follows directly from using (35), which accounts for the $k$ dependence of $\mathcal{R}_k$, while the remaining terms arise due to the $k$-dependence of $\hat{\phi}_k$, which therefore assume the form of an infinitesimal frame transformation. In Appendix A we give a more detailed derivation of (56) which generalises the derivation of the flow for the EAA presented in [13].

Now the question arises as to how $\Psi_k[\phi]$ should be determined. Evidently, we can arrive at a closed flow equation for $\Gamma_k[\phi]$ by specifying $\Psi_k[\phi]$ to be determined by $\Gamma_k[\phi]$ in some explicit manner. This is the approach pursued in other works [37, 38] in order to describe bound states through flowing bosonisation and exploited in [39–42] to describe hadronisation in QCD. The alternative, which we shall pursue, is instead to specify renormalisation conditions that constrain the form of $\Gamma_k[\phi]$ by fixing the values of the inessential couplings and solve the flow equation for the essential couplings and for parameters appearing in $\Psi_k[\phi]$ to determine the form of the frame transformation.

Let us note that, if we wish to impose a symmetry on $\Gamma[\phi]$ under some transformation of $\phi$ such as $\phi \to -\phi$, then one should impose that $\Psi_k[\phi]$ transforms in the same way as $\phi$. This requirement grants that the RG flow preserves the symmetry of the theory. Thus, if we want that $\Gamma_k[-\phi] = \Gamma_k[\phi]$, we should then impose that $\Psi_k[-\phi] = -\Psi_k[\phi]$.

As a final comment, let us now consider the limits $k \to 0$ and $k = \Lambda \to \infty$. In the limit $k \to 0$ the regulator $R_k(x_1, x_2)$ vanishes and thus we recover the 1PI effective action $\Gamma_0[\phi] = \Gamma[\phi]$ where $\hat{\phi}[\hat{\chi}] = \hat{\phi}_0[\hat{\chi}]$. In the opposite limit instead, making reference to (13), we can identify $M_\Lambda(x_1, x_2)$ by

$$\mathcal{R}_\Lambda(x_1, x_2) \sim M_\Lambda(x_1, x_2) . \tag{58}$$

Thus, $\Gamma_{k=\Lambda}[\phi] \sim S_{\hat{\phi}_\infty}[\phi]$ where $S_{\hat{\phi}_\infty}$ is given by (24). After giving an initial condition for the flow at $k = \Lambda$, the flow equation will then evolve towards the 1PI effective action while transforming the frame from $\hat{\phi}_\Lambda$ to $\hat{\phi}_0$.

### 3.2 Dimensionless covariant flow

In order to uncover RG fixed points, we need to work in units of the cutoff scale $k$ such that the RG flow, expressed in terms of dimensionless couplings $g_i$, obey an autonomous set of equations

$$\partial_t g_i = \beta_i(g) \,. \tag{59}$$

The passage to dimensionless variables can be done either by a passive frame transformation or by an active one. The active way, however, is more elegant and makes it also evident that the scale $k$ itself is simply an inessential coupling. To this end we define

$$\mathcal{N} \mathrm{e}^{-\Gamma_t[\varphi]} = \langle \mathrm{e}^{(\hat{\varphi}_t - \varphi) \cdot \frac{\delta}{\delta \varphi} \Gamma_t[\varphi] - \frac{1}{2}(\hat{\varphi}_t - \varphi) \cdot R \cdot (\hat{\varphi}_t - \varphi)} \rangle \,, \tag{60}$$

where we use $\varphi$ to denote the dimensionless fields and the subscript $t$ instead of $k$ to emphasise that there is no explicit dependence on $k$. In (60) the dimensionless regulator $R = R(\Delta)$ is understood as a function of the dimensionless Laplacian viewed as a two point function $\Delta(y_1, y_2) := -\partial_{y_1}^2 \delta(y_1 - y_2)$ where $y_1$ and $y_2$ are dimensionless coordinates. The expectation values of observables are given by

$$\langle \hat{\mathcal{O}} \rangle_{\varphi,t} = \mathrm{e}^{\Gamma_t[\varphi]} \langle \mathrm{e}^{(\hat{\varphi}_t - \varphi) \cdot \frac{\delta}{\delta \varphi} \Gamma_t[\varphi] - \frac{1}{2}(\hat{\varphi}_t - \varphi) \cdot R \cdot (\hat{\varphi}_t - \varphi)} \hat{\mathcal{O}} \rangle \,. \tag{61}$$

It is convenient to introduce the generator of dilatations $\psi_{\mathrm{dil}}$ as

$$\psi_{\mathrm{dil}}(y) := -y_\mu \partial_\mu \varphi(y) - \frac{d-2}{2} \varphi(y) \,, \tag{62}$$

in which the first term accounts for the rescaling of the coordinates and the second accounts for the rescaling of the field. In particular, if we have a term $\Xi[\varphi] = O(\varphi^n, \partial^s)$ in the action, such that $\Xi[\varphi]$ has canonical dimension $n(d-2)/2 + s - d$, one can show that

$$\psi_{\mathrm{dil}} \cdot \frac{\delta}{\delta \varphi} \Xi[\varphi] = -\left( n(d-2)/2 + s - d \right) \Xi[\varphi] \,. \tag{63}$$

In Appendix B we give the derivation of this equation. By defining the dimensionless RG kernel $\psi_t$ as

$$\psi_t^{\mathrm{tot}}[\varphi] := \psi_t[\varphi] + \psi_{\mathrm{dil}}[\varphi] := \langle \partial_t \hat{\varphi}_t[\hat{\chi}] \rangle_{\varphi,t} \,, \tag{64}$$

where $\psi_t^{\mathrm{tot}}$ denotes the total dimensionless RG kernel incorporating the dilatation step of the RG transformation, the dimensionless flow equation is given by

$$\left( \partial_t + \psi_t^{\mathrm{tot}}[\varphi] \cdot \frac{\delta}{\delta \varphi} \right) \Gamma_t[\varphi] = \mathrm{Tr} \frac{1}{\Gamma_t^{(2)}[\varphi] + R} \cdot \frac{\delta}{\delta \varphi} \psi_t^{\mathrm{tot}}[\varphi] \cdot R \,. \tag{65}$$

The form of (65) makes it clear that an RG transformation is nothing but an active frame transformation which includes a dilatation step where the conjugate inessential coupling is $k$ itself. This is inline with the observations made in [43] that show a direct relation between the flow of EAA and the anomaly due to the breaking of scale invariance.

To arrive at a more familiar form of the trace, we notice that the following identity holds

$$\mathrm{Tr} \frac{1}{\Gamma_t^{(2)}[\varphi] + R} \cdot \frac{\delta}{\delta \varphi} \psi_{\mathrm{dil}}[\varphi] \cdot R = \frac{1}{2} \mathrm{Tr} \frac{1}{\Gamma_t^{(2)}[\varphi] + R} \cdot \dot{R} \,, \tag{66}$$

where

$$\dot{R}(\Delta) := 2(R(\Delta) - \Delta R'(\Delta)) = \partial_t \mathcal{R}_k|_{k=1} \,, \tag{67}$$

which we prove in Appendix B. Using (66), it is then straightforward to show that (65) is (56) recast in dimensionless variables. In particular, the passive transformation (48) is given by

$$\hat{\varphi}(y) = k^{-(d-2)/2}\hat{\phi}(k^{-1}y) =: (c_{\text{dil}} \cdot \hat{\phi})(y), \tag{68}$$

and thus $c_{\text{dil}}(y, x_1) = k^{-(d-2)/2}\delta(k^{-1}y - x_1)$. The form of (62) then results from differentiating (68). Finally, let us then denote a dimensionless redundant operator by

$$\zeta\frac{\partial}{\partial\zeta}\Gamma_t = \mathcal{T}(\Gamma_t)\Phi[\varphi] := \Phi[\varphi] \cdot \frac{\delta}{\delta\varphi}\Gamma_t[\varphi] - \text{Tr}\,\frac{1}{\Gamma_t^{(2)}[\varphi] + R}\frac{\delta}{\delta\varphi}\Phi[\varphi] \cdot R, \tag{69}$$

where $\mathcal{T}(\Gamma_t)$ is understood as a $\Gamma_t$-dependent linear operator which acts on $\Phi[\varphi]$. Then the flow equation can be concisely written as

$$-\partial_t\Gamma_t[\varphi] = \mathcal{T}(\Gamma_t)(\psi_t[\varphi] + \psi_{\text{dil}}[\varphi]). \tag{70}$$

This form makes it explicit that the RG flow is simply a frame transformation.

## 3.3 Relation to Wilsonian flows

Let us end this Section by making contact with generalised flow equations for the Wilsonian effective action. If we relax the constraints on $\mathcal{R}_k$ such that we no longer view it as a regulator, one can obtain the flow equations for the Wilsonian effective action $S_k$ by taking the limit $\mathcal{R}_k \to \infty$. In particular, replacing the $\mathcal{R}_k \to \alpha\mathcal{R}_k$ and taking $\alpha \to \infty$ while denoting $\Gamma_k[\phi] \to S_k[\phi]$, the generalised flow equation (56) reduces to

$$\left(\partial_t + \Psi_k[\phi] \cdot \frac{\delta}{\delta\phi}\right)S_k[\phi] = \text{Tr}\,\frac{\delta}{\delta\phi}\Psi_k[\phi], \tag{71}$$

apart from a vacuum term which we neglect, while a redundant operator is given by

$$\zeta\frac{\partial}{\partial\zeta}S_k[\phi] = \Phi \cdot \frac{\delta}{\delta\phi}S_k[\phi] - \text{Tr}\,\frac{\delta}{\delta\phi}\Phi[\phi]. \tag{72}$$

These are the expressions for the generalised flow equation and redundant operators first written down in [15]. The reason we obtain the flow for the Wilsonian effective action in the limit $\mathcal{R}_k \to \infty$ is simple: this is due to the fact that the regulator term induces a delta function in the functional integral such that $\Gamma_{\hat{\phi}_k}[\phi, K] \to S_{\hat{\phi}_k}[\phi]$.

The flow equation (71) has been used to demonstrate scheme independence to different degrees [20–23]. However, in the flow equation (71), one has to introduce a UV-cuff into $\Psi_k[\phi]$ in order to regularise the trace. One advantage of the flow equations (56) is that the regulator $\mathcal{R}_k$ is disentangled from the RG kernel $\Psi_k[\phi]$, meaning that the trace will be regularised for any $\Psi_k[\phi]$ provided $\mathcal{R}_k$ decreases fast enough in the large momentum limit.

## 4 The standard scheme

### 4.1 Wetterich-Morris flow

As an example, in this Section, we focus on the simple case where one eliminates only a single inessential coupling, namely the wavefunction renormalisation $Z_k$ which is conjugate

to the redundant operator $\mathcal{T}(\Gamma_k)\varphi$. The removal of $Z_k$ then introduces the anomalous dimension of the field,

$$\eta_k = -\partial_t \log(Z_k),\tag{73}$$

and it is a necessary step to uncover fixed points with a non-zero anomalous dimension. As with the transition to dimensionless variables, $Z_k$ can be eliminated by an active frame transformation or by a passive transformation. By either method, we arrive at the Wetterich-Morris equation in the presence of a non-zero anomalous dimension [13,14]. By the active method, this is achieved by simply setting

$$\Psi_k[\phi] = -\frac{1}{2}\eta_k\phi,\tag{74}$$

from which we can infer that

$$\hat{\phi}_k = Z_k^{1/2}\hat{\phi}_0,\tag{75}$$

where we choose to impose $Z_0 = 1$ as the boundary condition. Following the passive route instead, we begin with the EAA $\Gamma_{\hat{\phi}_0,k}[\phi_0] = \Gamma[\phi_0, Z_k\mathcal{R}_k]$ which is given explicitly by

$$\mathcal{N}\mathrm{e}^{-\Gamma_{\hat{\phi}_0,k}[\phi_0]} = \left\langle \mathrm{e}^{(\hat{\phi}_0-\chi_0)\cdot\frac{\delta}{\delta\phi_0}\Gamma_{\hat{\phi}_0,k}[\phi_0]+\frac{Z_k}{2}(\hat{\phi}_0-\chi_0)\cdot\mathcal{R}_k\cdot(\hat{\phi}_0-\chi_0)}\right\rangle.\tag{76}$$

The flow equation is now given by

$$\partial_t\Gamma_{\hat{\phi}_0,k}[\phi_0] = \frac{1}{2}\mathrm{Tr}\frac{1}{\Gamma_{\hat{\phi}_0,k}^{(2)}[\phi_0] + Z_k\mathcal{R}_k}\cdot\partial_t(Z_k\mathcal{R}_k),\tag{77}$$

which is the standard form of the Wetterich-Morris equation, apart from making the dependence on the wavefunction renormalisation explicit. Then we make the passive change of frames (48) to eliminate $Z_k$ from the flow equation by setting $\phi_0 = Z_k^{-1/2}\phi$, where (49) implies that $\Gamma_k[\phi] = \Gamma_{\hat{\phi}_0,k}[Z_k^{-1/2}\phi]$. The flow equation (77) can then be recast in the form

$$\left(\partial_t - \frac{1}{2}\eta_k\phi\cdot\frac{\delta}{\delta\phi}\right)\Gamma_k[\phi] = \frac{1}{2}\mathrm{Tr}\,\mathcal{G}_k[\phi]\cdot(\partial_t\mathcal{R}_k - \eta_k\mathcal{R}_k),\tag{78}$$

which is now manifestly independent of $Z_k$ and is equal to (56) with $\Psi_k$ given by (74). The fact that the terms proportional to $\eta_k$ in (78) have the form of a redundant coupling then simply reflects the fact that $Z_k$ was inessential. In dimensionless variables the flow equation (78) is given by (65) where $\psi_t = -\frac{1}{2}\eta_k\varphi$.

## 4.2 Renormalisation conditions

We have arrived at the flow equation (78) without having specified the inessential coupling $Z_k$. This means that we have the freedom to impose a renormalisation condition that constrains the form of $\Gamma_k[\phi]$ by fixing the value of one coupling to some fixed value. Solving the flow equation (78) under the chosen renormalisation then determines $\eta_k$ as a function of the remaining couplings. In terms of $\Gamma_{\hat{\phi}_0,k}[\phi_0]$, this is equivalent to identifying $Z_k$ with one coupling. A typical choice is to expand the $\Gamma_{\hat{\phi}_0,k}[\phi_0]$ in fields and in derivatives and then identify $Z_k$ with the coefficient of the term $\frac{1}{2}\int_x(\partial_\mu\phi_0)(\partial_\mu\phi_0)$. In terms of $\Gamma_k[\phi]$ this fixes the coefficient of $\int_x(\partial_\mu\phi)(\partial_\mu\phi)$ to be 1/2. However, this choice is not unique. One can instead expand $\Gamma_k[\phi]$ only in derivatives such that

$$\Gamma_k[\phi] = \int_x\left[V_k(\phi) + \frac{1}{2}z_k(\phi)(\partial_\mu\phi)(\partial_\mu\phi)\right] + O(\partial^4),\tag{79}$$

where $V_k(\phi)$ and $z_k(\phi)$ are functions of the field and then choose the renormalisation condition

$$z_k(\tilde{\phi}) = 1\,, \tag{80}$$

for a single constant value of the field $\phi(x) = \tilde{\phi}$. The essential scheme which we present in the next sections is based on renormalisation conditions that generalise (80).

Before arriving at this generalisation, let us first scrutinise the choice (80) for the renormalisation condition to trace the reasoning behind it. To this end we note that $z_k(\tilde{\phi})$ is the inessential coupling conjugate to the redundant operator (69) in the case where $\Phi = \frac{1}{2}\varphi$, as it is clear from (78), namely

$$\frac{1}{2}\mathcal{T}(\Gamma_t)\varphi = \frac{1}{2}\varphi \cdot \frac{\delta}{\delta\varphi}\Gamma_t[\varphi] - \frac{1}{2}\operatorname{Tr}\mathcal{G}_t[\varphi] \cdot R\,. \tag{81}$$

In general, the redundant operator is a complicated functional of $\varphi$ since it depends on the form of $\Gamma_t[\varphi]$. However, at the Gaussian fixed point $\Gamma_t = \mathcal{K}$ with

$$\mathcal{K}[\varphi] := \frac{1}{2}\int_y (\partial_\mu\varphi)(\partial_\mu\varphi)\,, \tag{82}$$

one has that (81) reduces to the free action itself

$$\frac{1}{2}\mathcal{T}(\mathcal{K})\varphi = \frac{1}{2}\int_y (\partial_\mu\varphi)(\partial_\mu\varphi) + \text{constant}\,, \tag{83}$$

apart from a vacuum term. The fact that $\mathcal{K}$ is invariant under shifts $\varphi(y) \to \tilde{\varphi} + \varphi(y)$ then reveals why we were free to choose the renormalisation point $\tilde{\varphi}$. Thus any of the renormalisation conditions (80) will fix the same inessential coupling at the Gaussian fixed point. As we elaborate on in Appendix C, one can also fix inessential couplings at an alternative free fixed point by imposing an alternative renormalisation condition to eliminate $Z_k$. This makes it clear that the renormalisation condition (80) is intimately related to the kinematics of the Gaussian fixed point (82). Here we are discussing only a single inessential coupling. However, in general there is an infinite number of inessential couplings and we would like to impose renormalisation conditions to eliminate all of them. We may then ask whether there is a practical way to do so. In the next Section, we will present the minimal essential scheme which achieves this aim.

# 5 Minimal essential scheme

Our aim in this Section is to find a scheme that imposes a renormalisation condition for each inessential coupling $\zeta_\alpha$ by fixing them to some prescribed values. In order to solve the flow equations when applying multiple renormalisation conditions, we allow $\psi_t$ to depend on a set of *gamma functions* $\{\gamma_\alpha\}$, where we must include one gamma function for each renormalisation condition. The gamma functions, along with the beta functions for the remaining running couplings, are then found to be functions of the remaining couplings. For example, instead of fixing $\psi_t = -\frac{1}{2}\eta_k\varphi$, as in the standard scheme where we apply a single renormalisation condition, we can instead choose $\psi_t = \gamma_1(t)\varphi + \gamma_2(t)\varphi^3$ and then impose two renormalisation conditions which fixes the values of two inessential couplings. Solving the flow equation under these conditions, the gamma functions will then be determined as functions of the remaining running couplings. In general, we can write

$$\psi_t[\varphi] = \sum_\alpha \gamma_\alpha(t)\Phi_\alpha[\varphi]\,, \tag{84}$$

where the $\{\Phi_\alpha[\varphi]\}$ are a set of linearly independent local operators, one for each renormalisation condition which we impose. In essential schemes we include all possible local operators in the set $\{\Phi_\alpha[\varphi]\}$. Applying a renormalisation condition for each $\Phi_\alpha$ would then fix the value of all inessential couplings. For this purpose, we wish to find a practical set of renormalisation conditions that generalise the one applied in the standard scheme. Following the logic of the last Section, we therefore choose the renormalisation conditions such that we fix the values of the inessential couplings at the Gaussian fixed point. Inserting $\Gamma_t = \mathcal{K}$ into (69), the redundant operators at the Gaussian fixed point are given by

$$\mathcal{T}(\mathcal{K})\Phi_\alpha = \Phi_\alpha \cdot \Delta\varphi - \text{Tr}\, \frac{R}{\Delta + R} \cdot \frac{\delta}{\delta\varphi} \Phi_\alpha[\varphi]\,. \tag{85}$$

Then, in the minimal essential scheme we write the action such that it depends only on the essential couplings $\lambda$ by specifying the ansatz[5]

$$\Gamma_t[\varphi] = \mathcal{K} + \sum_a \lambda_a(t) e_a[\varphi]\,, \tag{86}$$

where $\{e_a[\varphi]\}$ are a set of operators which are linearly independent of the redundant operators (85) and together with the latter form a complete basis. Without loss of generality we can assume that the couplings behave as $\lambda_a(t) = e^{-\theta_G t}\lambda_a(0) + \dots$ in the vicinity of the Gaussian fixed point, in which case $e_a[\varphi]$ are the *scaling operators* at the Gaussian fixed point, $\theta_G$ the corresponding Gaussian critical exponents and the essential couplings $\lambda_a(t)$ are called the *scaling fields* in the literature [15].

The task of distinguishing the scaling operators from redundant operators at the Gaussian fixed point is made simpler by the following observation: if $\Phi_\alpha$ is a homogeneous function of the field of degree $n$, then the first term in (85) is a homogeneous function of degree $n + 1$, while the second term is a homogeneous function of degree $n - 1$. It follows from this structure that if $\{e_a[\varphi]\}$ are a set of operators which are linearly independent of $\Phi_\alpha \cdot \Delta\varphi$, they will also be linearly independent of $\mathcal{T}(\mathcal{K})\Phi_\alpha$. In other words, when identifying the scaling operators at the Gaussian fixed point, we can neglect the second term in (85) which is understood as a loop correction. To see this clearly, let us first assume that the scaling operators $e_a[\varphi]$ are linearly independent of $\Phi_\alpha \cdot \Delta\varphi$ such that

$$\sum_\alpha c_\alpha \Phi_\alpha \cdot \Delta\varphi + \sum_a c_a e_a[\varphi] = 0\,, \tag{87}$$

if and only if $c_\alpha = 0$ and $c_a = 0$. Then we can expand the redundant operator as

$$\mathcal{T}(\mathcal{K})\Phi_\alpha = \sum_\beta \tilde{\Upsilon}_{\alpha\beta} \Phi_\beta[\varphi] \cdot \Delta\varphi + \sum_a \tilde{v}_{\alpha a} e_a[\varphi]\,, \tag{88}$$

where $\tilde{\Upsilon}_{\alpha\beta}$ and $\tilde{v}_{\alpha a}$ are numerical coefficients. Then one can show that the eigenvalues of the matrix with components $\tilde{\Upsilon}_{\alpha\beta}$ will all be equal to one and thus $\tilde{\Upsilon}$ is an invertible matrix. To see that the eigenvalues of $\tilde{\Upsilon}$ are all equal to one, let's first consider the simple example where $\{\Phi_\alpha\} = \{\Phi_1, \Phi_2\} = \{\varphi, \varphi^3\}$ for which $\Upsilon$ has the form

$$\Upsilon = \begin{pmatrix} 1 & 0 \\ \tilde{\Upsilon}_{21} & 1 \end{pmatrix}\,, \tag{89}$$

where $\Upsilon_{21}$ is in general non-zero. The zero component follows from the fact that $\mathcal{T}(\mathcal{K})\varphi$ is linear in the field and therefore involves no term of the form $\varphi^3 \cdot \Delta\varphi$. The form of the matrix

---

[5]Here we neglect the vacuum energy term since it is independent of $\varphi$.

$\tilde{\Upsilon}$ is preserved in the general case by working in the basis where $\{\Phi_\alpha\} = \{\Phi_{\alpha_0}, \Phi_{\alpha_1}, \dots\}$, with $\alpha_n$ labelling each linearly independent local operator with $n$ powers of the field. For $n = 1$ we have $\Phi_{\alpha_1} = \{\varphi, \Delta\varphi, \dots\}$, while for $n = 2$ we have $\Phi_{\alpha_2} = \{\varphi^2, \varphi\Delta\varphi, (\partial_\mu\varphi)^2, \dots\}$, with the ellipses denoting terms involving four or more derivatives. Then the matrix $\Upsilon$ has the form

$$\tilde{\Upsilon} = \begin{pmatrix} 1 & 0 & 0 & \cdots \\ \tilde{\Upsilon}_{21} & 1 & 0 & \cdots \\ \tilde{\Upsilon}_{31} & \tilde{\Upsilon}_{32} & 1 & \cdots \\ \vdots & \vdots & \vdots & \ddots \end{pmatrix}, \tag{90}$$

which has all eigenvalues equal to one.

Having set the renormalisation conditions at the Gaussian fixed point, we know that the couplings $\lambda_a$ will be the essential couplings in the vicinity of the Gaussian fixed point. However, away from the Gaussian fixed point, the form of the redundant operators will change. Expanding the redundant operators for a general action of the form (86) we will obtain

$$\mathcal{T}(\Gamma_t)\Phi_\alpha[\varphi] = \sum_\beta \Upsilon_{\alpha\beta}(\lambda)\Phi_\beta[\varphi] \cdot \Delta\varphi + \sum_b v_{\alpha b}(\lambda)e_b[\varphi], \tag{91}$$

where $\Upsilon_{\alpha\beta}(\lambda)$ and $v_{\alpha b}(\lambda)$ are functions of the essential couplings and reduce to $\Upsilon_{\alpha\beta}(0) = \tilde{\Upsilon}_{\alpha\beta}$ and $v_{\alpha b}(0) = \tilde{v}_{\alpha b}$ at the Gaussian fixed point. At any point where $\Upsilon_{\alpha\beta}(\lambda)$ is invertible, the operators $R(\Gamma_t)\Phi_\alpha[\varphi]$ and $e_b[\varphi]$ will be linearly independent. The points for which $\Upsilon$ is not invertible form a disconnected hyper-surface consisting of all points in the essential theory space (i.e. the space spanned by the essential couplings $\lambda_a$), where

$$\det \Upsilon(\lambda) = 0. \tag{92}$$

On the hyper-surface (92), the flow will typically be singular. Therefore, adopting the minimal essential scheme puts a restriction on which physical theories we can have access to. However, it is intuitively clear that this restriction has a physical meaning since the theories in question are those that share the kinematics of the Gaussian fixed point. Indeed, a remarkable consequence of the minimal essential scheme is that the propagator evaluated at any constant value of the parameterised field $\varphi(x) = \tilde{\varphi}$ will be given by

$$\mathcal{G}_t[\tilde{\varphi}] = \frac{1}{q^2 + v_t^{(2)}(\tilde{\varphi}) + R(q^2)}, \tag{93}$$

where $v_t^{(2)}(\tilde{\varphi})$ is the second derivative of a dimensionless potential. This simple form follows since by integration by parts $\int_x (\varphi - \tilde{\varphi})\Delta^{s/2}(\varphi - \tilde{\varphi}) = \int_x \varphi\Delta^{s/2}\varphi$ for even integers $s \geq 2$. Let us hasten to point out that this does not imply that the propagator for the physical field $\hat{\chi}$ is of this form, but only that the propagator can be brought into this form by a frame transformation. In particular, the form (93) does not exclude the possibility that $\hat{\chi}$ develops an anomalous dimension $\eta$, namely that the connected two-point function of $\hat{\chi}$ scales as $\sim p^{-2+\eta}$.

# 6 Fixed points

In the vicinity of fixed points one can obtain universal scaling exponents which are independent of the renormalisation conditions which define different schemes. However, there are also critical exponents associated with redundant operators which are entirely scheme dependent. In this Section we will contrast features of essential schemes with those of the standard scheme in these respects.

### 6.1 Fixed points and scaling exponents

Fixed points of the exact RG are uncovered by looking at $t$-independent solutions of (65) such that the fixed point action $\Gamma_\star$ obeys

$$\left(\psi_\star^{\text{tot}}[\varphi] \cdot \frac{\delta}{\delta\varphi}\right)\Gamma_\star[\varphi] = \text{Tr}\frac{1}{\Gamma_\star^{(2)}[\varphi] + R} \cdot \frac{\delta}{\delta\varphi}\psi_\star^{\text{tot}}[\varphi] \cdot R\,, \tag{94}$$

which in general defines a relationship between $\psi_\star$ and $\Gamma_\star$.

The critical exponents associated with the fixed point are then found by perturbing the fixed point solution $\Gamma_\star$ by adding a small perturbation $\delta\Gamma_t = \Gamma_t - \Gamma_\star$ and similarly perturbing $\psi_\star$ by

$$\delta\psi_t = \left.\frac{\delta\psi_t}{\delta\Gamma_t}\right|_{\Gamma_t=\Gamma_\star}\delta\Gamma_t\,, \tag{95}$$

and studying the linearised flow equation for $\delta\Gamma_t$ which is given by

$$-\partial_t\delta\Gamma_t = \left(\frac{\delta\mathcal{T}(\Gamma_\star)}{\delta\Gamma_t}\psi_\star^{\text{tot}}\right)\delta\Gamma_t + \mathcal{T}(\Gamma_\star)\delta\psi_t\,. \tag{96}$$

The critical exponents $\theta$ are then defined by looking for eigenperturbations which are of the form

$$\delta\Gamma_t = \epsilon\,e^{-t\theta}\mathcal{O}[\varphi]\,, \quad \delta\psi_t = \epsilon\,e^{-t\theta}\Omega[\varphi]\,, \tag{97}$$

where $\mathcal{O}[\varphi]$ and $\Omega[\varphi]$ are $t$-independent. Depending on the sign of $\theta$, one refers to the operator $\mathcal{O}[\varphi]$ as *relevant* ($\theta > 0$), *irrelevant* ($\theta < 0$) or *marginal* ($\theta = 0$). We note that the functional form of $\mathcal{O}[\varphi]$ will depend on the frame and hence on the scheme. Physically, we know however that they must be the expectation value of the same observable $\hat{\mathcal{O}}$. Wegner [15] has shown that eigenperturbations fall into two classes: redundant eigenperturbations where $\mathcal{O}[\varphi]$ is a redundant operator, and therefore multiplied by an inessential coupling, and scaling operators which are linearly independent of the former (i.e. the analogs of $e_a[\varphi]$). At the Gaussian fixed point, the redundant operators are some linear combination of the redundant operators (85). More generally, the redundant operators at any fixed point, which have the form

$$\mathcal{O}_\Phi[\varphi] = \mathcal{T}(\Gamma_\star)\Phi[\varphi]\,, \tag{98}$$

have critical exponents $\theta$ which are entirely scheme dependent. Redundant eigenperturbations carry no physics and should be disregarded. Conversely, the scaling operators have scheme independent universal scaling exponents and are physical perturbations of the fixed point.

In the standard scheme, one removes only a single inessential coupling and thus one will have an infinite number of redundant eigenperturbations which must be disregarded. In essential schemes instead, all inessential couplings are removed and thus we automatically disregard all redundant eigenperturbations.

### 6.2 The redundant perturbation due to shifts

Actually, there remains one redundant operator which is not automatically disregarded in the minimal essential scheme, namely the one for which $\Phi[\varphi] = 1$. The reason for this is that the Gaussian action is invariant under constant shifts of the field $\varphi \rightarrow \varphi + \text{constant}$. Happily, this redundant operator can be treated exactly and hence it is nonetheless simple to disregard it. In fact, it is straightforward to show that $\mathcal{O}_{\text{shift}}[\varphi] := \mathcal{O}_{\Phi=1}[\varphi]$ is always

an eigenperturbation independently of the scheme, where

$$\mathcal{O}_{\text{shift}}[\varphi] = 1 \cdot \frac{\delta}{\delta\varphi} \Gamma_\star[\varphi], \tag{99a}$$

$$\Omega_{\text{shift}}[\varphi] = 1 \cdot \frac{\delta}{\delta\varphi} \psi_\star[\varphi] + \theta - \frac{d-2}{2}. \tag{99b}$$

To see that this will always be an eigenoperator, we can replace the field in the fixed point equation by $\varphi \to \varphi + \epsilon$ and expand to first order in $\epsilon$. This gives an identity obeyed by the fixed point action from which the solution (99) to the linearised flow follows immediately. In the standard scheme where $\psi_t[\varphi] = -\eta_k \frac{1}{2}\varphi$ it follows directly from (99b) that $\theta = \frac{d-2+\eta_\star}{2}$. In the minimal essential scheme, in order to fully determine $\psi_t[\varphi]$, we can impose that

$$\psi_t[0] = 0, \tag{100}$$

and then determine $\theta$ by setting $\varphi = 0$ in (99b). One then obtains

$$\theta = -1 \cdot \frac{\delta}{\delta\varphi} \psi_\star[\varphi] + \frac{d-2}{2}\bigg|_{\varphi=0}. \tag{101}$$

However (100) is only one choice and it is clear that by imposing a different condition, $\theta$ can take any value.

## 6.3 The anomalous dimension

Let us now discuss a scaling operator associated with the anomalous dimension. In the standard scheme, one introduces the parameter $\eta_k$ via the choice of the RG kernel. At a fixed point $\eta_k = \eta_\star = \eta$ is the anomalous dimension where we use $\eta$ to represent the universal critical exponent rather than $\eta_\star$ which is a parameter introduced in the RG kernel *only* in the standard scheme. The fact that $\eta = \eta_\star$ is the value of the universal exponent comes about because in the standard scheme there is a scaling relation between $\eta_\star$ and the scaling exponent for the operator $\mathcal{O} = \int_x \varphi$. To see this, we note that given a solution $\Gamma_k[\phi]$ to the flow equation (78), the EAA defined as $\Gamma_k[\phi] + Z_k^{-1/2} \int_x h\phi$ is still a solution to (78), provided $h$ is independent of $k$ and $\phi$. It is then evident that $h$ is nothing but a physical external field that couples to $\hat{\chi}$ in the microscopic action. At a fixed point, this means that there is always an eigenperturbation of this form. In dimensionless variables, the eigenperturbation is given by

$$\delta\Gamma_t = \epsilon\, e^{-t\frac{d+2-\eta_\star}{2}} \int_y \varphi, \tag{102}$$

and thus we see there is a scaling exponent given by $\theta = \frac{d+2-\eta_\star}{2}$. Thus, along with the other scaling exponents, $\theta = \frac{d+2-\eta_\star}{2}$ will be a universal quantity. However the simple form $\mathcal{O}[\varphi] = \int_x \varphi$ originates from the simple linear relation between $\hat{\phi}$ and $\hat{\chi}$ typical of the standard scheme and from the fact that in any frame a physical source must couple to one and the same field $\hat{\chi}[\hat{\phi}]$. In a general scheme, the relation between $\hat{\phi}$ and $\hat{\chi}$ will be non-linear and hence to compute $\eta$ we must instead look for an eigenperturbation of the form

$$\delta\Gamma_t = \epsilon \int_y \langle c_{\text{dil}} \cdot \hat{\chi}\rangle_{\varphi,t} \equiv \epsilon\, e^{-t\frac{d+2-\eta}{2}} \int_y \chi[\varphi], \tag{103}$$

where $\chi[\varphi] = \varphi$ only in the frame associated with the standard scheme. If we impose a symmetry on the fixed point action under $\varphi \to -\varphi$ then we will have that $\chi[-\varphi] = -\chi[\varphi]$. Apart from this characteristic, there is nothing that distinguishes $\frac{d+2-\eta}{2}$ from any other

747 scaling exponent. Thus to compute η we must look at odd eigenperturbations of an even
748 fixed point action. A related point, that has been recognised in [44], is that while $\eta_k$
749 approaches the particular value η at a fixed point, independently of the renormalisation
750 condition, this is not true of the gamma functions appearing in $\psi_t$ whenever $\psi_t$ is non-
751 linear.

# 7  The minimal essential scheme at order $\partial^2$

753 We will now derive the flow equation in the minimal essential scheme at order $\partial^2$ in the
754 derivative expansion. This is achieved by expanding the action as in (79) and neglecting
755 the higher derivative terms. However, in the minimal essential scheme the renormalisation
756 condition (80) is generalised such that

$$z_k(\phi) = 1 \,, \tag{104}$$

757 for *all* values of the field and all scales $k$. Thus, we go from fixing a single coupling in the
758 standard scheme to fixing a whole function of the field in the essential one. To close the
759 flow equations under this renormalisation condition, we set the RG kernel to

$$\Psi_k[\phi] = F_k(\phi(x)) \,, \tag{105}$$

760 where $F_k(\phi(x))$ is a function of the fields (without derivatives) constrained such that we
761 can solve the flow equation under the renormalisation condition (104). Therefore, working
762 at order $\partial^2$ the ansatz for the EAA is simply given by

$$\Gamma_k[\phi] = \int_x \left[ V_k(\phi) + \frac{1}{2}(\partial_\mu \phi)(\partial_\mu \phi) \right]. \tag{106}$$

763 Inserting (106) and (105) into (56) the l.h.s. is given by

$$\partial_t \Gamma_k[\phi] + \int_x \frac{\delta \Gamma_k[\varphi]}{\delta \phi(x)} F_k(\phi(x)) = \int_x \left[ \partial_t V_k(\phi) + F_k(\phi) V_k^{(1)}(\phi) + F_k^{(1)}(\phi) \, (\partial_\mu \phi) \, (\partial_\mu \phi) \right] , \tag{107}$$

764 where the super-script $(n)$ on functions of the field denotes their $n$-th derivative. These
765 terms depend on $F_k(\phi)$ and thus, instead of solving for $\partial_t V_k(\phi)$ and $\partial_t z_k(\phi)$, we will instead
766 solve for $\partial_t V_k(\phi)$ and $F_k(\phi)$. To find the equations for $\partial_t V_k$ and $F_k$, in Appendix D we
767 expand the trace on the r.h.s. of the flow equation (56) with the action given by (106)
768 and field renormalisation (105) up to order $\partial^2$. The result is given by

$$\partial_t V_k = -F_k \, V_k^{(1)} + \frac{1}{2(4\pi)^{d/2}} Q_{d/2} \left[ G_k \left( \partial_t \mathcal{R}_k + 2 F_k^{(1)} \mathcal{R}_k \right) \right] , \tag{108a}$$

$$
\begin{aligned}
F_k^{(1)} = & \frac{\left( V_k^{(3)} \right)^2}{2(4\pi)^{d/2}} \, Q_{d/2} \left[ G_k^2 G_k' \left( \partial_t \mathcal{R}_k + 2 F_k^{(1)} \mathcal{R}_k \right) \right] \\
& + \frac{\left( V_k^{(3)} \right)^2}{2(4\pi)^{d/2}} \, Q_{d/2+1} \left[ G_k^2 G_k'' \left( \partial_t \mathcal{R}_k + 2 F_k^{(1)} \mathcal{R}_k \right) \right] \\
& - \frac{V_k^{(3)} F_k^{(2)}}{(4\pi)^{d/2}} \left( Q_{d/2} \left[ G_k G_k' \mathcal{R}_k \right] + Q_{d/2+1} \left[ G_k G_k'' \mathcal{R}_k \right] \right) ,
\end{aligned}
\tag{108b}
$$

where we introduced the following quantities

$$P_k(z) = z + \mathcal{R}_k(z) \,, \tag{109}$$

$$G_k = \left( P_k + V_k^{(2)} \right)^{-1} \,, \tag{110}$$

$$Q_n[W] = \frac{1}{\Gamma(n)} \int_0^\infty \mathrm{d}z \, z^{n-1} \, W(z) \,. \tag{111}$$

The primes on $G_k$ indicate derivatives with respect to the momentum squared $z$.

# 8  Wilson-Fisher Fixed point

Let us now exemplify the minimal essential scheme at order $\partial^2$ by studying the 3D Ising model in the vicinity of the Wilson-Fisher fixed point.

## 8.1  Flow equations in $d = 3$

To this end, we specialise the study of Eqs. (108) to the case $d = 3$. In the following, we make use of the cutoff function [45]

$$\mathcal{R}_k(z) = (k^2 - z)\Theta(k^2 - z) \,, \tag{112}$$

where $\Theta(k^2-z)$ is the Heaviside theta function. This choice of the cutoff function leads to a particularly simple closed form of Eqs. (108). Being interested in critical scaling solutions of the RG flow, we transition to dimensionless variables such that the dimensionless field is given by $\varphi = k^{-\frac{1}{2}}\phi$ and the dimensionless functions are defined by $v = k^{-3}V$ and $f = k^{-\frac{1}{2}}F$. The equations (108) then read

$$\partial_t v_t(\varphi) + 3v_t(\varphi) - \frac{1}{2}\left[\varphi - 2f_t(\varphi)\right]v_t^{(1)}(\varphi) = b\,\frac{1 + \frac{2}{5}f_t^{(1)}(\varphi)}{1 + v_t^{(2)}(\varphi)} \,, \tag{113a}$$

$$-f_t^{(1)}(\varphi) = \frac{b}{2}\,\frac{\left[v_t^{(3)}(\varphi)\right]^2}{\left[1 + v_t^{(2)}(\varphi)\right]^4} \,. \tag{113b}$$

The constant $b$ takes the value $b = 1/(6\pi^2)$, however we note that $b$ can also be set to any positive real value $b \to \kappa^2 b$ since this is equivalent to performing the redefinitions $v_t(\varphi) \to v_t(\kappa\varphi)/\kappa^2$, $f_t(\varphi) \to f_t(\kappa\varphi)/\kappa$ and then rescaling the field by $\varphi \to \varphi/\kappa$. Choosing $b$ to take other values can be useful for numerical purposes, however, all our results are presented for $b = 1/(6\pi^2)$. Let us stress at this point that equations (113) have a simpler form as compared to the analogous equations [46] in the standard scheme using (112). In particular, in the minimal essential scheme, the $Q$-functionals (111) are simple rational functions of $v^{(2)}$ and $v^{(3)}$, whereas in the standard scheme they involve transcendental functions.

## 8.2  Scaling solutions

In the minimal essential scheme, scaling solutions are given by $k$-independent solutions $v(\varphi)$ and $f(\varphi)$ to Eqs. (113), which therefore solve the following system of ordinary dif-

794 ferential equations

$$3v(\varphi) - \frac{1}{2}\varphi\, v^{(1)}(\varphi) + f(\varphi)v^{(1)}(\varphi) = b\,\frac{1 + \frac{2}{5}f^{(1)}(\varphi)}{1 + v^{(2)}(\varphi)}\,, \tag{114a}$$

$$-f^{(1)}(\varphi) = \frac{b}{2}\,\frac{\left[v^{(3)}(\varphi)\right]^2}{\left[1 + v^{(2)}(\varphi)\right]^4}\,. \tag{114b}$$

795 We notice that differentiating the first equation w.r.t. $\varphi$, yields an equation for $v^{(3)}$
796 which is expressed in terms of lower derivatives of $v$ and $f$. Once this expression for $v^{(3)}$ is
797 substituted into the second equation, the system reduces to a second-order differential one.
798 The so-obtained equation for $f$ turns out to be quadratic in $f^{(2)}$. Solving algebraically
799 for $f^{(2)}$ we therefore have two roots. We thus conclude that any solution of (114) can be
800 characterised by a set of four initial conditions along with the choice of one of the roots.
801 We are interested in globally-defined solutions $v(\varphi) = v_\star(\varphi)$ and $f(\varphi) = f_\star(\varphi)$ to (114)
802 which are well-defined for all values of $\varphi \in \mathbb{R}$. These solutions correspond to fixed points of
803 the RG. Furthermore the $\mathbb{Z}_2$ symmetry of the Ising model demands that $v_\star(\varphi)$ and $f_\star(\varphi)$
804 should be even and odd functions respectively. Looking at the behaviour of any putative
805 fixed-point solution in the large-field limit one realises that if a globally-defined solution
806 exists, then for $\varphi \to \pm\infty$ it must behave as

$$v(\varphi) = A_V\,\varphi^6 + O(\varphi^5)\,, \tag{115}$$

$$f(\varphi) = \pm A_F + O(\varphi^{-9})\,, \tag{116}$$

807 with all the higher-order terms being determined as functions of $A_V$ and $A_F$. On the other
808 hand, to ensure the correct parity of the corresponding scaling solution, one finds that, by
809 studying the equations (114), it is necessary and sufficient to impose the conditions[6]

$$\{v^{(1)}(0) = 0,\ f^{(1)}(0) = 0\}\,, \tag{117}$$

810 which are obtained by expanding (114) around $\varphi = 0$. In particular, we notice that (117)
811 and (114) imply that $f(0) = 0$. Thus, the expansion at infinity gives us two free parameters
812 which must be chosen such that at $\varphi = 0$ the conditions (117) are met. We thus expect at
813 most a countable number of acceptable fixed point solutions to Eqs. (114). As expected
814 we have found only two, namely the Gaussian and the Wilson-Fisher fixed points.
815 In order to show this result, we can numerically solve the equations (114) for different
816 initial conditions at $\varphi = 0$. This is convenient since, by imposing (117), we are left with
817 only one boundary condition which we can take to be the dimensionless mass squared
818 $\sigma := v^{(2)}(0)$. In addition to $\sigma$ we also have to choose the root for $f^{(2)}$. The two roots can
819 be distinguished by noticing that in the limit $\sigma \to 0$, one root displays the Gaussian fixed
820 point while the other does not. By setting the initial conditions at $\varphi = 0$ we are therefore
821 left with two one-parameter families of solutions.
822 As the above reasoning dictates, one immediately realises that only a countable number
823 of solutions exist globally for all values of $\varphi \in \mathbb{R}$. Generic solutions which starts at $\varphi = 0$
824 end at a singularity located at a finite value of the field $\varphi = \varphi_s(\sigma)$. We can therefore plot
825 the function $\varphi_s(\sigma)$ to find those values $\sigma_\star$ for which $\varphi_s(\sigma)$ diverges: these are the values
826 for which the corresponding solution of Eqs. (114) is globally-defined. In Fig. 1 (top-left
827 panel) we show the result of this search for well-defined scaling solutions selecting the root
828 which possesses the Gaussian fixed point and scanning $\sigma$ in the range $-1 < \sigma < 0$. This
829 technique is sometimes referred to as *spike-plot* because globally well-defined solutions,

---

[6]Equivalently, the conditions $\{f(0) = 0,\ f^{(1)}(0) = 0\}$ imply that $v^{(1)}(0) = 0$.

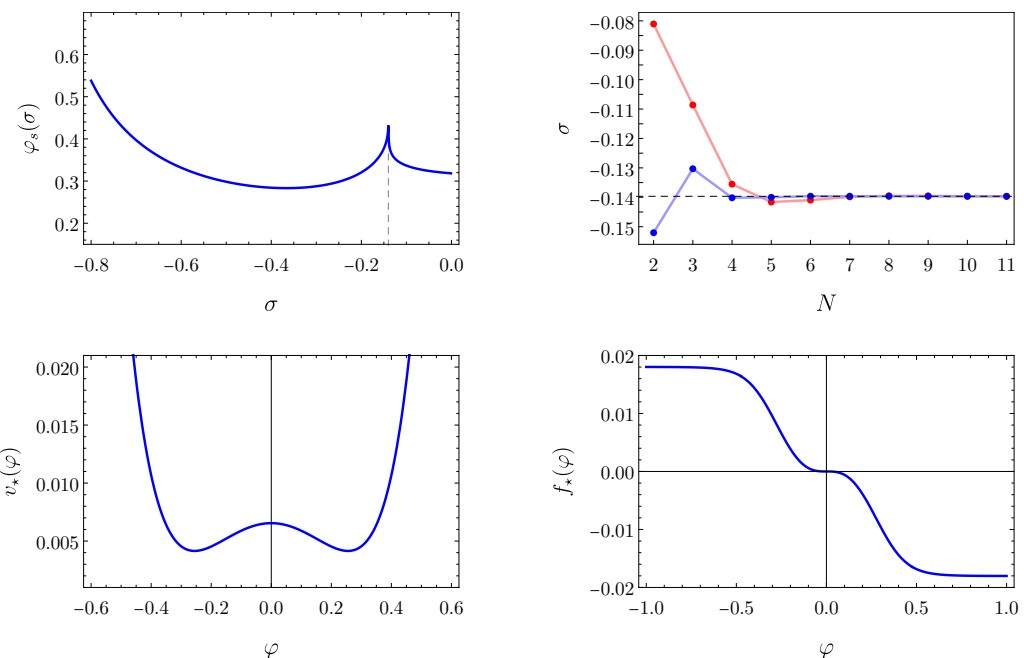

Figure 1: In the top-left panel, we show the singular values $\varphi_s(\sigma)$ as a function of $\sigma$. The spike located at $\sigma_\star = -0.13967$ represents the Wilson-Fisher universality class. The value of $\sigma_\star = v_\star^{(2)}(0)$ obtained from the expansion around $\rho = 0$ (red) and the expansion around the minimum $\bar{\rho}_\star$ (blue) as a function of the truncation order $N$ is showed in the top-right panel where the dashed line represents the corresponding functional value obtained from the spike-plot. The globally-defined fixed-point effective potential $v_\star(\varphi)$ and RG kernel $f_\star(\varphi)$ corresponding to the Wilson-Fisher fixed point solution are given in the bottom-left and bottom-right panels respectively.

namely divergences in $\varphi_s(\sigma)$, appear as spikes [25, 46–48]. The Wilson-Fisher fixed point solution is found at

$$\sigma_\star = -0.13967 \,. \tag{118}$$

In passing, we observe that the family of solutions which include the Gaussian fixed point also displays Wilson-Fisher fixed point, while we have detected no spike in the other family.

In order to corroborate the spike-plot analysis, we searched for scaling solutions by expanding $v_\star(\varphi)$ and $f_\star(\varphi)$ in powers of the fields up to a finite order $N$. For this purpose it is convenient to re-express $v_\star$ and $f_\star$ in terms of the manifest $\mathbb{Z}_2$ invariant $\rho(\varphi) \equiv \frac{1}{2}\varphi^2$. Expanding around $\rho = 0$ to order $N$ we can write $v$ and $f$ as

$$v_\star(\varphi) = \sum_{n=0}^{N} \lambda_{2n}^\star \rho^n \,, \tag{119a}$$

$$f_\star(\varphi) = \varphi \sum_{n=1}^{N-1} \gamma_{2n+1}^\star \rho^n \,, \tag{119b}$$

(such that $v_\star(\varphi)$ is even and $f_\star(\varphi)$ is odd), while expanding around the minimum $\bar{\rho}_\star =$

$\frac{1}{2}\varphi_{\min\star}^2$ of the fixed-point potential, our truncations are given by

$$v_\star(\varphi) = \bar\lambda_0^\star + \sum_{n=2}^{N} \bar\lambda_{2n}^\star \left(\rho - \bar\rho^\star\right)^n , \tag{120a}$$

$$f_\star(\varphi) = \varphi \sum_{n=0}^{N-1} \bar\gamma_{2n+1}^\star \left(\rho - \bar\rho^\star\right)^n . \tag{120b}$$

The equations (114), expanded in $\rho$ around $\rho = 0$ ($\rho = \bar\rho_\star$) reduce to algebraic equations for the couplings $\lambda_{2n\star}$ ($\bar\lambda_{2n\star}$ and $\bar\rho_\star$) and the fixed point values $\gamma_{2n\star}$ ($\bar\gamma$). Solving these algebraic solutions we find approximate scaling solutions at each order $N$ which converge, as $N$ is increased, to the corresponding scaling solution we obtained numerically from the spike-plot. In particular the values of $\sigma_\star = v_\star^{(2)}(0)$ found at each order $N$ in the two expansions is plotted in Fig. 1 (top-right panel) and are seen to converge to the functional value (118). We thus conclude that the approximate solutions at order $N$ converge to the globally-defined numerical solutions as $N \to \infty$.

We close this Section by a remark: in the spike-plot approach, the task of integrating the scaling equations to find a globally defined solution involves fine tuning $\sigma$. In practice, to obtain the global functions $v_\star(\varphi)$ and $f_\star(\varphi)$, we have taken advantage of the asymptotic solutions (115) and (116) and of the expansion around the minimum (120). Specifically, in order to determine values for $A_F$ and $A_V$ we can match the $v(\varphi)$ and $\frac{\partial v(\varphi)}{\partial \rho}$ for values of the field where the expansion around the minimum and the large field one overlap. This determines

$$A_V \approx 1.35 , \tag{121}$$

$$A_F \approx -0.018 . \tag{122}$$

Although the expansions of $f(\varphi)$ do not perfectly overlap, a suitable Padé approximant to the large field expansion eventually matches the expansion around the minimum. The corresponding globally-defined functions $v_\star(\varphi)$ and $f_\star(\varphi)$ at the Wilson-Fisher fixed point are plotted in the bottom panels of Fig. 1. An in-depth analysis of global fixed points and their relation to local expansions has been given in [49, 50].

## 8.3 Eigenperturbations

To obtain the critical exponents for the Wilson-Fisher fixed point we solve the flow equations (113) in the vicinity of the scaling solution. Functionally, perturbations of the scaling solution

$$\delta v_t(\varphi) = v_t(\varphi) - v_\star(\varphi) , \tag{123a}$$

$$\delta f_t(\varphi) = f_t(\varphi) - f_\star(\varphi) \tag{123b}$$

obey the linearised flow equation

$$\partial_t \delta v_t(\varphi) = \frac{1}{2}\left[\varphi - 2f_\star(\varphi)\right]\delta v_t^{(1)}(\varphi) - 3\delta v_t(\varphi) - v_\star^{(1)}(\varphi)\delta f_t(\varphi) + \frac{2b\,\delta f_t^{(1)}(\varphi)}{5\left[1 + v_\star^{(2)}(\varphi)\right]} +$$

$$- \frac{b\left[5 + 2f_\star^{(1)}(\varphi)\right]\delta v_t^{(2)}(\varphi)}{5\left[1 + v_\star^{(2)}(\varphi)\right]^2} , \tag{124a}$$

$$-\delta f_t^{(1)}(\varphi) = \frac{b\,v_\star^{(3)}(\varphi)\,\delta v_t^{(3)}(\varphi)}{\left[1 + v_\star^{(2)}(\varphi)\right]^4} - \frac{2b\left[v_\star^{(3)}(\varphi)\right]^2 \delta v_t^{(2)}(\varphi)}{\left[1 + v_\star^{(2)}(\varphi)\right]^5} . \tag{124b}$$

Similarly to the fixed point equations (114), these can be converted into second order differential equations. We note that, since $v_\star(\varphi)$ is an even function, and $f_\star(\varphi)$ is an odd function, one can consider even and odd perturbations $\delta v_t(\varphi)$ separately. In order to find the spectrum of scaling exponents $\theta_n$ we can express a general perturbation as a sum of its eigenperturbations[7]

$$\delta v_t(\varphi) = \sum_n C_n \mathrm{e}^{-\theta_n t} \mathcal{O}_n(\varphi) \, , \tag{125a}$$

$$\delta f_t(\varphi) = \sum_n C_n \mathrm{e}^{-\theta_n t} \Omega_n(\varphi) \, , \tag{125b}$$

where $C_n$ are undetermined constants that parameterise the perturbations of the fixed point and $n$ runs over the spectrum of eigenperturbations. For each $n$ the functions $\Psi_n$ and $\Omega_n$ obey a pair of coupled second order differential equations which depend on $\theta_n$. The sum is justified by the fact that the spectrum $\theta_n$ is quantised. To show this, first we consider the large field limit $\varphi \to \infty$ where we determine that

$$\mathcal{O}_n = A_n \varphi^{6-2\theta_n} + 6 \left( \theta_n - \frac{1}{2} \right)^{-1} A_V B_n \varphi^5 \dots \, , \tag{126}$$

$$\Omega_n = B_n + \dots \tag{127}$$

up to subleading terms. This introduces two parameters $A_n$ and $B_n$ for each eigenperturbation. Considering the behaviour around $\varphi = 0$, for even and odd perturbations we have that $\mathcal{O}_n^{(1)}(0) = 0$ and $\mathcal{O}_n(0) = 0$ respectively. Furthermore the linearity of the equations allows us to normalise even and odd perturbations by $\mathcal{O}_n(0) = 1$ and $\mathcal{O}_n^{(1)}(0) = 1$. Imposing that the RG kernel vanishes at vanishing field (100) then enforces that $\Omega_n(0) = 0$ for either parity. On the other hand $\Omega_n^{(1)}(0) = 0$ follows automatically from (124b) since $v_\star(\varphi)$ is even (and hence $v_\star^{(3)}(0) = 0$). Therefore we need to satisfy three independent boundary conditions at $\varphi = 0$ to ensure the correct parity, while we only have two free parameters $A_n$ and $B_n$. As a result, the allowed values of $\theta_n$ must be quantised to satisfy all three boundary conditions.

## 8.4 Scaling exponents

In order to compute the scaling exponents $\nu$ and $\omega$ we look at even eigenperturbations. Here we shall use $t$-dependent generalisations of the expansions (119) and (120) to compute the exponents at order $N$ in both expansions. The couplings $\lambda_{2n}$, $\bar{\lambda}_{2n}$ and $\bar{\rho}$ are now $k$-dependent with beta functions

$$\partial_t \lambda_{2n} = \beta_{2n}(\lambda) \, , \tag{128a}$$

$$\partial_t \bar{\lambda}_{2n} = \bar{\beta}_{2n}(\bar{\lambda}, \bar{\rho}) \, , \tag{128b}$$

$$\partial_t \bar{\rho} = \beta_{\bar{\rho}}(\bar{\lambda}, \bar{\rho}) \, , \tag{128c}$$

and similarly $\gamma_{2n} = \gamma_{2n}(\lambda)$ and $\bar{\gamma}_{2n} = \bar{\gamma}_{2n}(\bar{\lambda}, \bar{\rho})$ are also determined as functions of the couplings. The critical exponents obtained from the expansion around $\varphi = 0$ are obtained from eigenvalues of the stability matrix

$$M_{nm}^{\mathrm{even}} = \left. \frac{\partial \beta_{2n}}{\partial \lambda_{2m}} \right|_{\lambda = \lambda^\star} \, , \tag{129}$$

---

[7]This is a slight abuse of notation since earlier we denoted eigenperturbations of the fixed point action as $\mathcal{O}$ while $\mathcal{O}_n$ are perturbations of the fixed point potential.

where $\lambda_\star$ denotes the values of the couplings at the Wilson-Fisher fixed point. Similarly, by defining $\bar{\lambda}_2 := \bar{\rho}$ and $\bar{\beta}_2 := \beta_{\bar{\rho}}$, the stability matrix for the expansion around the minimum is defined by

$$\bar{M}_{nm}^{\text{even}} = \frac{\partial \bar{\beta}_{2n}}{\partial \bar{\lambda}_{2m}}\bigg|_{\bar{\lambda}=\bar{\lambda}^\star} . \tag{130}$$

The critical exponents are equal to minus the eigenvalues of the stability matrix. In particular, the critical exponent $-1/\nu$ is identified with the sole relevant eigenvalue (ignoring the vacuum energy), which has a negative real part, while the correction-to-scaling exponent $\omega$ is identified with the irrelevant eigenvalue with the smallest positive real part. The values of these exponents at different orders $N$ up to $N = 11$ are shown in Fig 2 (top-right and bottom-left panels). We observe that the critical exponents converge towards as the order $N$ is increased and in general the expansion around the minimum converges faster w.r.t. the one around zero. At order $N = 11$ in the expansion around the minimum we find that

$$\nu = 0.6271 , \tag{131}$$

$$\omega = 0.8350 . \tag{132}$$

In order to compute the scaling exponent $\eta$ we look at odd perturbations $\delta v_t(\varphi)$ and even perturbations $\delta f_t(\varphi)$. This introduces a set of beta functions for couplings that multiply odd functions of the field and which, though vanishing at the Wilson-Fisher fixed point, exhibit non-zero scaling exponents. These exponents have been computed in using the exact RG in [51].

These odd perturbations also include the redundant perturbation due to shifts (99). Imposing (100), which implies $\Omega_{\text{shift}}(0) = 0$, we then have that the critical exponent (101) is given by $\theta_{\text{shift}} = 1/2$ since $1 \cdot \frac{\delta}{\delta\varphi}\psi_\star[0] = f_\star^{(1)}(0) = 0$. Thus (99) reduces to $\mathcal{O}_{\text{shift}} = \int_x v_\star^{(1)}(\varphi)$ and $\Omega_{\text{shift}} = f_\star^{(1)}(\varphi)$. Of course there is nothing physical about the value $1/2$ since we can obtain any value for the scaling exponent $\theta_{\text{shift}}$ by instead considering the perturbation of $f_\star$ where $\Omega_{\text{shift}} = f_\star^{(1)}(\varphi) + c$ for any value of $c$ which leads to $\theta_{\text{shift}} = 1/2 + c$. This is equivalent to choosing a condition other than $f_t(0) = 0$. In any case, this redundant perturbation is easily identified and discarded.

To calculate the anomalous dimension $\eta$, we again use expansions around vanishing field and around the minimum of the potential $v_\star(\varphi)$. At order $N$ in the expansion around $\varphi = 0$, we expand $\delta v_t(\varphi)$ and $\delta f_t(\varphi)$ as

$$\delta v_t(\varphi) = \varphi \sum_{n=0}^{N-1} \lambda_{2n+1}\rho^n , \tag{133a}$$

$$\delta f_t(\varphi) = \varphi^2 \sum_{n=0}^{N-1} \gamma_{2n+2}\rho^n , \tag{133b}$$

while the expansion around the minimum is written as

$$\delta v_t(\varphi) = \varphi \sum_{n=0}^{N-1} \bar{\lambda}_{2n+1}\left(\frac{1}{2}\varphi^2 - \bar{\rho}^\star\right)^n , \tag{134a}$$

$$\delta f_t(\varphi) = \varphi^2 \sum_{n=0}^{N-1} \bar{\gamma}_{2n+2}\left(\frac{1}{2}\varphi^2 - \bar{\rho}^\star\right)^n , \tag{134b}$$

and we notice that these expansions ensure that the boundary condition (100) is satisfied. With these forms of the perturbations, the linearised equations (124) are odd. One can then factor out a power of $\varphi$ to obtain even equations which can be expanded in the $\mathbb{Z}_2$

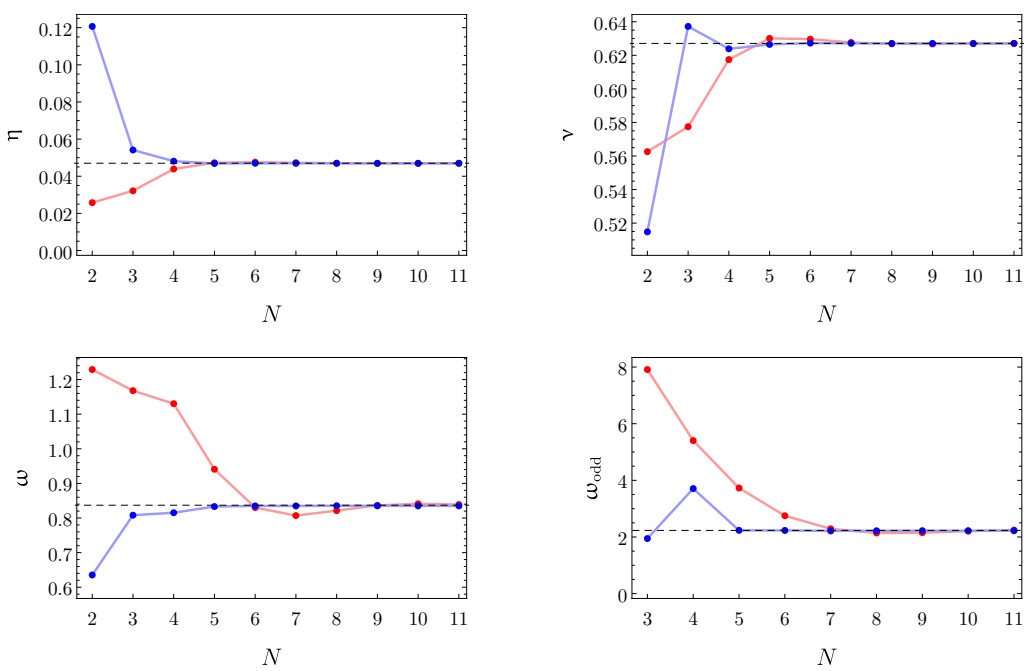

Figure 2: Critical exponents $\eta$ (top-left), $\nu$ (top-right), $\omega$ (bottom-left), $\omega_{\mathrm{odd}}$ (bottom-right), as a function of the truncation order $N$ for the expansions around $\rho = 0$ (red) and the expansion around the minimum of the potential $\bar{\rho}$ (blue). Dashed lines represent the numerical values given in the main text.

invariant $\rho$ around $\rho = 0$ and $\bar{\rho}^\star$. The linearised equations expanded around $\rho = 0$ ($\rho = \bar{\rho}^\star$) can then be solved for $\beta_{2n+1}$ and $\gamma_{2n+2}$ which are both linear in $\lambda_{2n+1}$. We then obtain the critical exponents from the stability matrices

$$M_{nm}^{\mathrm{odd}} = \left.\frac{\partial \beta_{2n+1}}{\partial \lambda_{2m+1}}\right|_{\lambda = \lambda^\star}, \tag{135a}$$

$$\bar{M}_{nm}^{\mathrm{odd}} = \left.\frac{\partial \bar{\beta}_{2n+1}}{\partial \bar{\lambda}_{2m+1}}\right|_{\lambda = \lambda^\star}, \tag{135b}$$

at each order $N$ in the two expansions. In the spectrum of odd eigenperturbations we find a single relevant positive critical exponents (disregarding $\theta_{\mathrm{shift}}$) which we identify as $(5 - \eta)/2$ in accordance with (103). As with $\nu$ and $\omega$ we find that the numerical value of $\eta$ converges $N \to \infty$. The values of $\eta$ at orders $N = 2$ to $N = 11$ are plotted in the top-left panel of Fig. 2. At order $N = 11$ we find

$$\eta = 0.0470. \tag{136}$$

We have also confirmed that this value $\eta$ is independent of the boundary condition (100). The convergence of the least irrelevant eigenvalue $\omega_{\mathrm{odd}} = -\theta$ associated to an odd perturbation shows a slower convergence than $\eta$. At order $N = 11$ in the expansion around the minimum the first three digits have converged to

$$\omega_{\mathrm{odd}} = 2.22. \tag{137}$$

As a remark, we notice here that at the specific values of $N = 3$ ($N = 4$), the exponents $\omega$ ($\omega_{\mathrm{odd}}$) are complex. One can also consider solving the linearised equations for perturbations with both even and odd parts obtaining a stability matrix from which $\nu$, $\omega$, $\eta$ and $\omega_{\mathrm{odd}}$ can all be obtained with the same values obtained from treating the perturbations separately.

## 9  Higher orders of derivative expansion

Having demonstrated the minimal essential scheme at order $\partial^2$, let us now discuss how it can be generalised to higher orders in the derivative expansion. Within the standard scheme, the EAA $\Gamma_k$ at order $\partial^4$ in the derivative expansion can be expressed as [30–32]

$$\Gamma_k = \int_x \left\{ V_k(\rho) + \frac{1}{2} z_k(\rho) \left( \partial_\mu \phi \, \partial_\mu \phi \right) + \right.$$
$$\left. + W_k^a(\rho) \left( \Delta \phi \right)^2 + W_k^b(\rho) \phi \Delta \phi \left( \partial_\mu \phi \, \partial_\mu \phi \right) + W_k^c(\rho) \left( \partial_\mu \phi \, \partial_\mu \phi \right)^2 \right\}, \tag{138}$$

where the three functions $W_k^i(\rho)$, with $i = a, b, c$ are linearly independent with respect to integration by parts.

We notice that both $W_k^a(\rho)$ and $W_k^b(\rho)$ are in the form of $\Phi \cdot \Delta \phi$, and hence in the minimal essential scheme the EAA reduces to

$$\Gamma_k = \int_x \left\{ V_k(\rho) + \frac{1}{2} \left( \partial_\mu \phi \, \partial_\mu \phi \right) + W_k(\rho) \left( \partial_\mu \phi \, \partial_\mu \phi \right)^2 \right\}, \tag{139}$$

which involves only two functions, namely the effective potential $V_k(\rho)$ and $W_k(\rho) \equiv W_k^c(\rho)$. In order to cope with the essential program, we generalise the RG kernel (105) to allow for terms involving up to two derivatives, namely

$$\Psi_k(x) = F_0(\phi) + F_{2,a}(\phi) \Delta \phi + \phi F_{2,b}(\phi) \left( \partial_\mu \phi \, \partial_\mu \phi \right). \tag{140}$$

Inserting the ansatz (139) into the l.h.s. of the flow equation (56), we note that this produces all of the terms at fourth order in the derivative expansion, namely

$$\partial_t \Gamma_k + \int_x \frac{\delta \Gamma_k}{\delta \phi} \Psi_k = \int_x \left\{ \partial_t V_k + F_0 V_k^{(1)} + \left[ F_0^{(1)} + V_k^{(1)} \phi F_{2,b} + \left( V_k^{(1)} F_{2,a} \right)^{(1)} \right] \left( \partial_\mu \phi \, \partial_\mu \phi \right) + \right.$$
$$\left. + F_{2,a} \left( \Delta \phi \right)^2 + \phi F_{2,b} \Delta \phi \left( \partial_\mu \phi \, \partial_\mu \phi \right) + \left[ \partial_t W_k + F_0 W_k^{(1)} + 4 W_k F_0^{(1)} \right] \left( \partial_\mu \phi \, \partial_\mu \phi \right)^2 \right\} + O(\partial^6). \tag{141}$$

It is easy to generalise this procedure to higher orders in derivative expansion. For example, at order $\partial^6$ we have to include all possible terms up to four derivatives in the RG kernel

$$\Psi_k(x) = F_0 + F_{2,a} \Delta \phi + \phi F_{2,b} \left( \partial_\mu \phi \, \partial_\mu \phi \right) + F_{4,a} \Delta^2 \phi + F_{4,b} \left( \Delta \phi \right)^2 + F_{4,c} \Delta \phi \left( \partial_\mu \phi \, \partial_\mu \phi \right)$$
$$+ F_{4,d} \left( \partial_\mu \phi \, \partial_\mu \phi \right)^2 + F_{4,e} \left( \partial_\mu \Delta \phi \right) \left( \partial_\mu \phi \right). \tag{142}$$

In this way, we reduce the number of operators in the ansatz for the EAA from 13 to 5. In the following table we show the comparison between the number of operators for $\Gamma_k$ in the standard and essential schemes.

|  | standard | essential |
|---|---|---|
| LPA | 1 | 1 |
| $\partial^2$ | 2 | 1 |
| $\partial^4$ | 5 | 2 |
| $\partial^6$ | 13 | 5 |
| $\vdots$ | $\vdots$ | $\vdots$ |

While at order $s = 0$ (i.e. in the LPA) the minimal essential scheme coincides with the standard scheme, the essential one can be carried out at any order in the derivative expansion, reducing its complexity order by order. At a given order $\partial^s$, the procedure of minimal essential scheme can be summarised as follows

⋄ Apart from the canonical kinetic term with coefficient $1/2$, eliminate all operators of the form $\Phi \cdot \Delta \phi$ from the ansatz of $\Gamma_k$;

⋄ insert all the possible terms up to order $\partial^{(s-2)}$ into the RG kernel $\Psi_k(x)$;

⋄ use equation (56) to find a set of beta functions for the essential operators which remain in the EAA, plus a set of equations which determine the functions appearing in the RG kernel $\Psi_k$.

Note that the final number of equations which one must solve at each order of the derivative expansion is the same as in the standard scheme. However, in the minimal essential scheme we obtain beta functions only for the essential couplings. Moreover, since the ansatz for EAA becomes simpler in the minimal essential scheme, the complexity in the calculation of the fluctuation contribution is reduced. In particular, the simple form of the propagator (93) evaluated at a constant field configuration is guaranteed.

# 10    Discussion

As we have both elucidated and demonstrated, the fact that the values of the inessential couplings are arbitrary can be used to one's advantage in practical QFT computations. This is made possible within the exact RG by the exact flow equation (56), derived by allowing the field variables $\hat{\phi}_k$ to themselves depend on the renormalisation scale $k$. This then allows us to solve the flow equation in a scheme where we provide a renormalisation condition for every inessential coupling. In these essential schemes, one only has to compute the flow of essential couplings. This has the advantage that the flow of inessential couplings, which cannot carry any physical information and therefore can only distract us from the physics, is automatically disregarded. The focus of this paper has been on the minimal essential scheme applied to a single scalar field and we have explicitly worked out the details for the derivative expansion. It is clear that these advantages are not restricted to this narrow scope. As such, here we take the opportunity to adopt a broader view of essential schemes and discuss their possible applications.

## 10.1    Non-minimal essential schemes and extended PMS studies

In the minimal essential scheme which we have presented, one sets all inessential couplings to zero apart from the coefficient of the kinetic term, which is fixed to be equal to one half. The motivation of this particular essential scheme is to minimise the complexity of calculations. It is in this sense that the minimal essential scheme is minimal, with the most striking simplification being the minimal form of the propagator (93). However, this choice of scheme is just one possibility and it can be that there are other useful schemes where the inessential couplings take non-trivial values. One possibility is instead to look for optimised schemes by applying the principle of minimal sensitivity to a given observable computed in a given approximation. In general terms, the PMS states that optimised schemes are those for which the inessential couplings take the values $\zeta = \zeta_{\text{PMS}}$ for which

$$\frac{\partial}{\partial \zeta} (\text{observable}) \bigg|_{\zeta = \zeta_{\text{PMS}}} = 0 \, . \tag{143}$$

This being the case for all values of $\zeta$ only if the observable is computed without making an approximation. In practice, however, there will be a discrete set of values of $\zeta_{\text{PMS}}$ for which (143) is satisfied.

It is natural to look for optimised schemes by considering non-minimal variants of the minimal essential scheme, where we continue to specify the values of all inessential couplings but relax the requirement that they take trivial values. In particular, we are free to write the general ansatz

$$\Gamma_t[\varphi] = \sum_a \lambda_a(t)e_a[\varphi] + \Phi_t[\varphi] \cdot \Delta\varphi\,, \tag{144}$$

where

$$\Phi_t[\varphi] = \sum_\alpha \zeta_\alpha \Phi_\alpha[\varphi] = \frac{1}{2}z_t(\varphi) + O(\partial^2)\,. \tag{145}$$

We thus reintroduce the inessential couplings $\zeta_\alpha$ which parameterise $\Phi_t[\varphi]$.[8] To close the flow equation without introducing independent beta functions for the inessential couplings one can set

$$\zeta_\alpha = \zeta_\alpha(\lambda)\,, \tag{146}$$

where the functions $\zeta_\alpha(\lambda)$ are prescribed functions of the essential couplings. With the restriction that $\Phi_t[\varphi] = \mathcal{K}$ when $\lambda = 0$, such that we still have the Gaussian fixed point in the canonical form[9], we are otherwise largely free to pick the functions $\zeta_\alpha(\lambda)$. Different prescriptions which specify every inessential coupling are *non-minimal essential schemes*. At order $\partial^2$ in the derivative expansion non-minimal essential schemes correspond to solving two flow equations which depend on three functions $v_t(\varphi)$, $z_t(\varphi)$, and $f_t(\varphi)$ by choosing $z_t(\varphi)$ to be completely determined by the potential $v_t(\varphi)$.

Although the complexity of calculations is increased with respect to the minimal essential scheme one can look for optimised schemes by applying the PMS. For example, one can study the dependence of the universal scaling exponents at a non-trivial fixed point to determine values $\zeta_\alpha(\lambda_\star) = \zeta_\alpha^{\text{PMS}}$ which satisfy the PMS criteria

$$\frac{\partial}{\partial \zeta_\alpha(\lambda_\star)}\theta(\zeta^{\text{PMS}}) = 0\,. \tag{147}$$

Since there is an infinite number of inessential couplings, we can in principle attempt to locate an extremum (147) in an infinite-dimensional space. In practice we can vary a finite number of the inessential couplings for example by letting $z_t(\varphi) = z_\star(\varphi) + O((\lambda - \lambda_\star)^2)$ and choosing $z_\star(\varphi)$ to be a finite order polynomial. It is therefore possible to make extended field-dependent PMS studies which are not possible in the standard scheme. This may lead to a better determination of physical quantities at a fixed order in the derivative expansion than those obtained in the standard scheme [30]. Thus a natural next step in the application of essential schemes is to perform an extended PMS study of the Ising critical exponents at order $\partial^2$.

## 10.2  Redundancies and symmetries

As well as arriving at a practical scheme for the exact RG our work also clarifies some important conceptual points. In particular, regarding the existence of redundant operators, it is abundantly clear that there is one redundant operator for each inessential coupling.

---

[8]Here we are making a slight abuse of notation since we have not properly identified $\lambda_a$ and $\zeta_\alpha$ as essential and inessential couplings respectively. We ignore these subtleties for the purpose of this discussion.

[9]One can, of course, choose a non-canonical form of the Gaussian fixed point but there would seem no particular practical advantage in doing so.

F. Wegner has proved by linearising the flow equations around a given fixed point, the inessential couplings do not appear in the linearised beta functions of the essential couplings [15]. Physically, we know it must be true since it is this property that ensures that universal scaling exponents are independent of the unphysical inessential couplings. The underlying mathematical reason is that there is a symmetry associated with each inessential coupling which together form a group (the group of frame transformations) that has closed Lie algebra. However, when making approximations, this property may be lost if the symmetries are broken and therefore a spurious dependence on the inessential couplings may arise. In particular, if this property does not hold, the criteria that an operator be an eigenperturbation and a redundant operator will seemingly overconstrain the eigenvalue problem [52]. To see this clearly, imagine we have one essential coupling $\lambda$ and one inessential coupling $\zeta$ obeying the following system of linearised beta functions $\partial_t \lambda = M_{\lambda\lambda}\lambda + M_{\lambda\zeta}\zeta$ and $\partial_t \zeta = M_{\zeta\lambda}\lambda + M_{\zeta\zeta}\zeta$. Then if $M_{\lambda\zeta} = 0$, it is clear that the redundant operator conjugate to $\zeta$ is an eigenperturbation since letting $\zeta$ be non-zero does not cause $\lambda$ to run. On the other hand, if in an approximation $M_{\lambda\zeta} \neq 0$, then the redundant operator will not be an eigenperturbation. This can then lead one to conclude that redundant eigenperturbations are rare since there must be a symmetry in order to satisfy both criteria. However, this apparent rareness is an artefact of making approximations, since it is the closed nature of the Lie algebra associated with frame invariance that provides the required infinite number of symmetries independently of the scheme. In an essential scheme, this problem is avoided by fiat since the redundant perturbations are disregarded. It may be fruitful nonetheless to find approximation schemes that preserve frame covariance, such that physical quantities are scheme independent at each order of the approximation scheme. Some progress in this direction has been made at second order of the derivative expansion for a variant of the Wilsonian effective action [53, 54].

## 10.3    Generalisability

The minimal essential scheme and the non-minimal variants can be straightforwardly generalised to theories with different field content, symmetries and the inclusion of fermionic fields. Given the many applications of the exact RG to a wide array of physical systems, we can expect that essential schemes can be useful both in reducing complexity and in order to find optimised schemes to compute observables. In particular, the application of essential schemes to gauge theories could reduce spurious dependence on gauge fixing parameters and background fields, since these are both examples of inessential couplings. Moreover, we mention here that essential schemes can possibly shed light on the issue of generalising the exact RG to problems involving boundaries. In particular, removing inessential coupling from the boundary action may help to preserve general boundary conditions along the RG flow.

## 10.4    Vertex expansion

Our focus in this paper has been on the simplifications that arise at each order in the derivative expansion, however, essential schemes can also be applied in other systematic approximation schemes. One such scheme is the vertex expansion where the EAA is expanded in terms of the $n$-point functions $\Gamma_k^{(n)}[0]$ to some finite order. If we approximate $\Gamma_k$ as depending on up to $N$ powers of the field then we should include up to $N-1$ powers of the field in $\Psi_k$ in order to solve the flow equation in an essential scheme. This can allow us to account for the full momentum dependence while keeping $N$ finite. For example, to ensure that the two-point function takes the simple form $-\partial^2 + m^2$ we should include a term $-\frac{1}{2}\eta_k(\Delta)\phi$ in $\Psi_k$ which accounts for the general linear field reparameterisation. In

fact, a scheme that removes all redundant operators from the two-point function in this manner has been put forward in [55]. The minimal essential scheme, applied consistently to a vertex expansion, would generalise this scheme by removing all redundant operators from the higher $n$-point functions include in the approximation.

## 10.5  Asymptotic Safety

Applying the minimal essential scheme to quantum gravity for example reduces the problem of finding a non-trivial fixed point underlying the asymptotic safety scenario [56]. Indeed this is the context in which Weinberg has suggested that such a scheme should be used [2]. Furthermore, a concrete proposal for a minimal scheme for quantum gravity has been put forward in [57]. While some works do utilise field redefinitions [58, 59], this has not been pursued at one-loop and at first order in the $\epsilon = d - 2$ expansion. For this purpose, essential schemes could be combined with the recently developed background independent and diffeomorphism invariant flow equation [60]. The fact that the propagator will take the simple form (93) is of special importance since this may guarantee that the theory is unitary and thus offer an answer to recent criticisms of the current asymptotic program [61]. More generally, by adopting the minimal essential scheme we are specifying a priori that the theory space that we are flowing is that of interacting particles whose kinematics are those of the Gaussian fixed point with two derivatives. This is a restriction on which fixed points we can find since, for example, we will not uncover fixed points associated with higher-derivative theories. However, we can expect that any fixed points that we do find will be unitary when we Wick rotate back to Lorentzian signature and reconstruct the propagator of the graviton [62].

## 10.6  Cosmology

In the context of scalar-tensor theories essential schemes could be used to resolve the cosmological frame equivalence question, building on recent progress [63–65]. In particular, adopting the principle of frame invariance ensures the physical equivalence of theories expressed in the Jordan and Einstein frames. Furthermore, one can apply renormalisation conditions to remain in the Einstein frame along the RG flow, where computations are typically easier, by generalising the minimal essential scheme.

# 11  Conclusion

Any description of Nature that we write down as a mathematical model will always depend on how we choose to parameterise or label physical objects (whether we make this decision consciously or not). On the other hand, Nature does not depend on how we label things; a rose by any other name would smell as sweet. However, taking the attitude that "any parameterisation will do" is not practical since solving a model is typically simpler by parameterising the physics in a particular way. A better attitude is to first identify which parameters of the model are inessential and tune them to simplify the task of solving the model. K. Wilson's exact renormalisation group embodies a complementary attitude to physics in which one does not write down a model but rather computes the model by solving a flow equation. In essential schemes, we adopt both attitudes such that we are not solving for the inessential couplings but only the for essential ones. In this way, what we solve for is not the mathematical model but only those physical quantities we are ultimately interested in. This distinction is very clear when we compute critical exponents at a critical point. In both the standard scheme and in essential schemes we will get a

spectrum of critical exponents. However, it is the spectrum of the latter that will only contain critical exponents which characterise a physical scaling law realised in Nature. As such, one should bear in mind that in the standard scheme not all critical exponents will be physical and that if we assume that they are, we can come to incorrect conclusions. In particular, there is nothing to prevent an inessential coupling to appear relevant in some schemes and therefore to give an incorrect counting of the number of relevant couplings at a non-trivial fixed point.

# Acknowledgements

We are indebted to B. Delamotte for motivating us during the early stages of the preparation of our manuscript. We then thank B. Delamotte and R. Percacci for a careful reading of the manuscript and for providing us with useful comments and suggestions.

**Funding information**    RBAZ acknowledges the support from the French ANR through the project NeqFluids (grant ANR-18-CE92-0019).

# A    Flow equation with general frame transformations

In this Appendix, we present a derivation of Eq. (56), which generalises the demonstration of the flow for the EAA presented in [13], and its development is strictly related to the classical derivation of the flow equation in the standard scheme  (78).
   Our scheme for the ERG is based on the idea that the basic degrees of freedom could flow along the RG trajectory. For this purpose, let us consider the generator of the connected correlation functions

$$\mathcal{W}_{\hat{\chi}}[J] \coloneqq \log \int (\mathrm{d}\hat{\chi}) \ \mathrm{e}^{-S_{\hat{\chi}}[\hat{\chi}] + \int_x J(x)\hat{\chi}(x)} , \tag{148}$$

where $J$ is an external source. We now introduce a scale dependent generalisation of Eq. (148) which depends on an IR cutoff scale $k$ by making two modifications. First we couple a source $J$ to a $k$-dependent field $\hat{\phi}_k[\hat{\chi}]$ which is a functional of the fundamental field $\hat{\chi}$. The new field $\hat{\phi}_k[\hat{\chi}]$ satisfies the following relations

$$\langle \hat{\phi}_k[\hat{\chi}] \rangle_{\phi,k} = \phi , \tag{149}$$

$$\langle \partial_t \hat{\phi}_k[\hat{\chi}] \rangle_{\phi,k} = \Psi_k[\phi] . \tag{150}$$

In a second step, we introduce an IR cutoff by adding the following term to the action

$$\Delta S_k[\hat{\phi}_k] = \frac{1}{2} \int_{x_1,x_2} \hat{\phi}_k(x_1) \mathcal{R}_k(x_1,x_2) \hat{\phi}_k(x_2) , \tag{151}$$

where $\mathcal{R}_k(x_1,x_2)$ is an IR cutoff function which can be chosen arbitrarily, provided it meets few constraints to ensure that the RG flow interpolates between the microscopic theory in the UV and the full effective theory in the IR. These modifications define the $k$-dependent generating functional

$$\mathrm{e}^{\mathcal{W}_{\hat{\phi}}[J]} \coloneqq \int (\mathrm{d}\hat{\chi}) \ \mathrm{e}^{-S_{\hat{\chi}}[\hat{\chi}] + \int_x J(x)\hat{\phi}_k(x) - \frac{1}{2} \int_{x_1,x_2} \hat{\phi}_k(x_1) \mathcal{R}_k(x_1,x_2) \hat{\phi}_k(x_2)} , \tag{152}$$

in terms of which the expectation values of arbitrary operators $\mathcal{O}$ can be obtained by differentiating the $\mathcal{W}_{\hat{\phi}}[J]$ as

$$
\begin{aligned}
\langle \hat{\mathcal{O}}[\hat{\phi}_k] \rangle &= e^{-\mathcal{W}_{\hat{\phi}}[J]} \hat{\mathcal{O}}[\delta/\delta J] e^{\mathcal{W}_{\hat{\phi}}[J]} \\
&= e^{-\mathcal{W}_{\hat{\phi}}[J]} \int (d\hat{\chi}) \; \hat{\mathcal{O}}[\hat{\phi}_k] e^{-S_{\hat{\chi}}[\hat{\chi}] + \int_x J(x)\hat{\phi}_k(x) - \frac{1}{2} \int_{x_1,x_2} \hat{\phi}_k(x_1)\mathcal{R}_k(x_1,x_2)\hat{\phi}_k(x_2)} \,.
\end{aligned} \tag{153}
$$

In particular, let's denote the $k$-dependent average (classical) field by

$$
\phi(x) = \frac{\delta}{\delta J(x)} \mathcal{W}_{\hat{\phi}}[J] \,, \tag{154}
$$

so that higher-order derivatives of $\mathcal{W}_{\hat{\phi}}$ are naturally related to correlation functions of $\hat{\phi}_k$. In this respect, the $k$-dependent connected two-point function can be defined as

$$
\mathcal{G}_k(x_1,x_2) \equiv \frac{\delta^2 \mathcal{W}_{\hat{\phi}}}{\delta J(x_1)\delta J(x_2)} = \langle \hat{\phi}_k(x_1)\hat{\phi}_k(x_2) \rangle - \phi(x_1)\phi(x_2) \,. \tag{155}
$$

We now seek a closed RG equation for $\mathcal{W}_{\hat{\phi}}[J]$. For a given choice of $\Psi_k[\phi]$, by differentiating Eq. (152) with respect to the RG time $t$ we obtain

$$
\begin{aligned}
\partial_t \mathcal{W}_{\hat{\phi}}[J] = \int_x \Psi_k[\phi(x)]J(x) &- \frac{1}{2} \int_{x_1,x_2} \langle \hat{\phi}_k(x_1)\hat{\phi}_k(x_2) \rangle \partial_t \mathcal{R}_k(x_1,x_2) \\
&- \int_{x_1,x_2} \langle \partial_t \hat{\phi}_k(x_1)\hat{\phi}_k(x_2) \rangle \mathcal{R}_k(x_1,x_2) \,.
\end{aligned} \tag{156}
$$

Using (154), differentiating Eq. (150) with respect to $J(x_2)$

$$
\begin{aligned}
-\phi(x_2)\Psi_k[\phi(x_1)] + \langle \partial_t \hat{\phi}_k(x_1)\hat{\phi}_k(x_2) \rangle &= \int_{x_3} \frac{\delta\phi(x_3)}{\delta J(x_2)} \frac{\delta\Psi_k[\phi(x_1)]}{\delta\phi(x_3)} \\
&= \int_{x_3} \frac{\delta^2 \mathcal{W}_{\hat{\phi}}[J]}{\delta J(x_2)\delta J(x_3)} \frac{\delta\Psi_k[\phi(x_1)]}{\delta\phi(x_3)} \,.
\end{aligned} \tag{157}
$$

Then we note that by taking advantage of the previous identity and using Eq. (155) we finally obtain the following closed flow equation

$$
\begin{aligned}
\partial_t \mathcal{W}_{\hat{\phi}}[J] = \int_x \Psi_k[\phi(x)]J(x) &- \frac{1}{2} \int_{x_1,x_2} \left[ \frac{\delta^2 \mathcal{W}_{\hat{\phi}}}{\delta J(x_1)\delta J(x_2)} + \phi(x_1)\phi(x_2) \right] \partial_t \mathcal{R}_k(x_1,x_2) \\
&- \int_{x_1,x_2} \left[ \phi(x_2)\Psi_k[\phi_k(x_1)] + \int_{x_3} \frac{\delta^2 \mathcal{W}_{\hat{\phi}}[J]}{\delta J(x_2)\delta J(x_3)} \frac{\delta\Psi_k[\phi(x_1)]}{\delta\phi(x_3)} \right] \mathcal{R}_k(x_1,x_2) \,.
\end{aligned} \tag{158}
$$

Let us now introduce the effective average action $\Gamma_k[\phi]$ by the following modified Legendre transformation

$$
\Gamma_k[\phi] = -\mathcal{W}_{\hat{\phi}}[J] + \int_x J(x)\phi(x) - \frac{1}{2} \int_{x_1,x_2} \phi(x_1)\mathcal{R}_k(x_1,x_2)\phi(x_2) \,, \tag{159}
$$

which is intended to be a functional of the average field such that

$$
\frac{\delta\Gamma_k[\phi]}{\delta\phi(x_1)} = J(x_1) - \int_x \mathcal{R}_k(x_1,x)\phi(x) \,. \tag{160}
$$

Differentiating Eq. (160) w.r.t. $\phi(x_2)$ and Eq. (154) w.r.t $J(x_1)$ yields the following identity

$$
\int_x \mathcal{G}_k(x_1,x)(\Gamma^{(2)} + \mathcal{R}_k)(x,x_2) = \delta(x_1 - x_2) \,. \tag{161}
$$

Taking advantage of Eqs. (160-161) and differentiating Eq. (159) with respect to $t$, the desired flow of $\Gamma_k[\phi]$ can be finally expressed as in Eq. (56), namely

$$\partial_t \Gamma_k[\phi] + \int_x \frac{\delta \Gamma_k[\phi]}{\delta \phi(x)} \Psi_k[\phi(x)] = \frac{1}{2} \int_{x_1,x_2} \frac{1}{\Gamma_k^{(2)} + \mathcal{R}_k}(x_1,x_2) \partial_t \mathcal{R}_k(x_2,x_1)$$
$$+ \int_{x_1,x_2,x_3} \frac{1}{\Gamma_k^{(2)} + \mathcal{R}_k}(x_1,x_2) \frac{\delta \Psi_k[\phi(x_3)]}{\delta \phi(x_2)} \mathcal{R}_k(x_3,x_1) \,. \quad (162)$$

One can also express $\Gamma_k[\phi]$ directly as the solution to integro-differential equation

$$\mathrm{e}^{-\Gamma_k[\phi]} = \int (\mathrm{d}\hat{\chi}) \; \mathrm{e}^{-S_{\hat{\chi}}[\hat{\chi}] + \int_x \frac{\delta \Gamma_k[\phi]}{\delta \phi}(\hat{\phi}_k(x) - \phi(x)) - \frac{1}{2} \int_{x_1,x_2} (\hat{\phi}_k(x_1) - \phi(x_1)) \mathcal{R}_k(x_1,x_2)(\hat{\phi}_k(x_2) - \phi(x_2))} \,.$$
$$(163)$$

In the paper we focus on the derivative expansion: this means that $\Psi_k[\phi]$ is given by Eq. (105) at order $O(\partial^2)$ , by Eq. (140) at order $O(\partial^4)$ and by Eq. (142) at order $O(\partial^6)$. Another possibility is to consider the vertex expansion, where $\Psi_k[\phi]$ is expressed in powers of the field with coefficients depending on the momenta

$$\Psi_k[\phi(x)] = \sum_n \int_{p_1,\dots,p_n} \Psi_k(p_1,\dots,p_n) \phi(p_1) \dots \phi(p_n) \, \mathrm{e}^{-\mathrm{i}x(p_1 + \dots + p_n)} \,. \quad (164)$$

# B    Properties of the dilatation operator

In this Appendix we present the main passages in order to demonstrate Eq. (63), which is related to $\psi_{\mathrm{dil}}$, and identity (66), needed to find the dimensionless version of the flow equation for EAA given in Eq. (70). Let us show that the term $-y^\mu \partial_\mu$ in $\psi_{\mathrm{dil}}$, given in (62), counts the number of derivatives. Denoting

$$\partial_r = \partial_{\mu_1} \dots \partial_{\mu_r} \,, \quad (165)$$

then if

$$\Phi[\varphi] = \Phi(\varphi(y), \partial_{\mu_1} \varphi(y), \dots) = O(\partial^s) \,, \quad (166)$$

such that

$$\Xi[\varphi] = \int_y \Phi[\varphi] \,, \quad (167)$$

we have that

$$\sum_r r \frac{\partial \Phi}{\partial \partial_r \varphi(x)} \partial_r \varphi(x) = s \Phi(x) \,. \quad (168)$$

Additionally we have that

$$[\partial_r, y^\mu \partial_\mu] = r \partial_r \,, \quad (169)$$

which can be proved by induction. Then using the above identities and integrating by parts we have that

$$y^\mu \partial_\mu \varphi \cdot \frac{\delta}{\delta \varphi} \int_y \Phi(y) = \int_y \sum_r \frac{\partial \Phi}{\partial \partial_r \varphi(y)} \partial_r y^\mu \partial_\mu \varphi(x)$$
$$= s \int_y \Phi + \int_y \sum_r \frac{\partial \Phi}{\partial \partial_r \varphi(y)} y^\mu \partial_\mu \partial_r \varphi(y)$$
$$= s \int_y \Phi + \int_y y^\mu \partial_\mu \Phi$$
$$= (s - d) \int_y \Phi \,. \quad (170)$$

Finally adding this contribution to the multiplicative contribution of $\psi_{\text{dil}}$ we obtain Eq. (63) . Let us now prove the identity (66)

$$\text{Tr} \frac{1}{\Gamma_t^{(2)}[\varphi] + R} \cdot \frac{\delta}{\delta\varphi} \psi_{\text{dil}}[\varphi] \cdot R = \frac{1}{2} \text{Tr} \frac{1}{\Gamma_t^{(2)}[\varphi] + R} \cdot \dot{R} \,. \tag{171}$$

In order to lighten the notation we drop the spacetime indexes, but it is clear that $\partial_y\, y = \partial_q\, q = d$ . Starting from the r.h.s. of identity (66) we have

$$\text{Tr} \frac{1}{\Gamma_t^{(2)}[\varphi] + R} \cdot \frac{\delta}{\delta\varphi} \psi_{\text{dil}}[\varphi] \cdot R = \int_{y_1,y_2,y_3} \mathcal{G}(y_1,y_2) \frac{\delta\psi_{\text{dil}}(y_3)}{\delta\phi(y_2)} R(y_3,y_1)$$

$$= \int_{y_1,y_2} \mathcal{G}(y_1,y_2)\, R(y_3,y_1) \left( -y_3\, \partial_{y_3} - \frac{d-2}{2} \right) \delta(y_3 - y_2)$$

$$= \int_{y_1,y_2} \mathcal{G}(y_1,y_2) \left( y_2\, \partial_{y_2} + d - \frac{d-2}{2} \right) R(y_2,y_1)$$

$$= \int_{y_1,y_2} \int_q \mathcal{G}(y_1,y_2) \left( -\mathrm{i}y_2\, q + \frac{d}{2} + 1 \right) R(q^2)\mathrm{e}^{-\mathrm{i}q(y_2-y_1)} \,. \tag{172}$$

Then we can rewrite the non trivial part of the previous expression as

$$\int_{y_1,y_2,q} \mathcal{G}(y_1,y_2)\,(\mathrm{i}y_2\, q)\, R(q^2)\mathrm{e}^{-\mathrm{i}q(y_2-y_1)} = \frac{1}{2} \int_{y_1,y_2,q} \mathcal{G}(y_1,y_2)\,\mathrm{i}\,(y_2 - y_1)\, q\, R(q^2)\mathrm{e}^{-\mathrm{i}q(y_2-y_1)} \tag{173}$$

$$= \frac{1}{2} \int_{y_1,y_2,q} \mathcal{G}(y_1,y_2)\, q\, R(q^2) \left( -\partial_q \mathrm{e}^{-\mathrm{i}q(y_2-y_1)} \right) \tag{174}$$

$$= \frac{1}{2} \int_{y_1,y_2,q} \mathcal{G}(y_1,y_2)\, \partial_q \left( q\, R(q^2) \right) \mathrm{e}^{-\mathrm{i}q(y_2-y_1)} \tag{175}$$

$$= \frac{1}{2} \int_{y_1,y_2,q} \mathcal{G}(y_1,y_2) \left[ d\, R(q^2) + q\, \partial_q R(q^2) \right] \mathrm{e}^{-\mathrm{i}q(y_2-y_1)} \,, \tag{176}$$

where in the first passage we just write $y_2$ as $(y_2 + y_2)/2$ and then in the second term we exchange $y_1$ and $y_2$ using the symmetry of the propagator and send $q \to -q$. So putting everything together

$$\int_{y_1,y_2,q} \mathcal{G}(y_1,y_2) \left( \mathrm{i}y_2\, q - \frac{d}{2} + 1 \right) R(q^2)\mathrm{e}^{-\mathrm{i}q(y_2-y_1)} = \int_{y_1,y_2,q} \mathcal{G}(y_1,y_2) \left( 1 - q^2\partial_{q^2} \right) R(q^2)\mathrm{e}^{-\mathrm{i}q(y_2-y_1)} \tag{177}$$

$$= \frac{1}{2} \text{Tr} \frac{1}{\Gamma_t^{(2)}[\varphi] + R} \cdot \dot{R} \,, \tag{178}$$

where $\dot{R}(\Delta) := 2[R(\Delta) - \Delta R'(\Delta)]$, given in Eq. (67).

# C  Renormalisation conditions in the standard scheme

In this Appendix, we discuss renormalisation conditions for the inessential coupling present in free theories. We have seen that in the standard case we impose Eq. (80) to fix the wave function renormalization but one can ask what happens for the high temperature

fixed point or higher-derivatives theories. Indeed, another renormalisation condition could be to fix one of the couplings appearing in the potential $V_k(\phi)$. For example we could fix

$$V_k^{(2)}(\phi_0) = Ck^2. \tag{179}$$

However these choices are not inconsequential since they can limit which fixed points can be found. In general terms a given fixed point solution $\Gamma_\star[\varphi]$ can be found only for a subset of all renormalisation conditions. In order to be able to find all fixed points one can instead choose to keep $\eta_\star$ arbitrary. A simple example is to look for free fixed points which can be treated exactly. In this case we can write (ignoring the vacuum term)

$$\Gamma_k[\phi] = \frac{1}{2}\phi \cdot k^2 H_k(-\partial^2/k^2) \cdot \phi, \tag{180}$$

where fixed points are solutions where $H_k(q^2) = H_\star(q^2)$ is independent of $k$. We arrive at the fixed point equation

$$q^2 \frac{\partial}{\partial q^2} H_\star(q^2) = \left(1 - \frac{1}{2}\eta_\star\right) H_\star(q^2). \tag{181}$$

If we impose that $H_\star(q^2)$ should be analytic around $q^2 = 0$ then the only solutions are $H_\star(q^2) = C\left(q^2\right)^{\frac{1}{2}s}$ where $\frac{1}{2}s$ is a non-negative integer given by $s = 2 - \eta_\star$ and thus the values that $\eta_\star$ can take is quantised and $C$ is an underdetermined number. In particular, for $s = 2$ the action is given by (79) with $V_k = 0$ and $z_k = C$, while for $s = 0$, which corresponds to the high temperature fixed point, we have $V_k = \frac{1}{2}k^2\phi^2$ and $z_k = 0$, with all higher derivative terms zero in both cases. This is of course a convoluted way to arrive at the conclusion that at free fixed points with $s$ derivatives the canonical dimension is given by $(d - s)/2$.

Now suppose we had chosen (80), then the only free fixed point that we could have found would be the one where $s = 2$. On the other hand if instead we had imposed (179), then we could only have found the high temperature fixed point where $s = 0$. Since the number $C$ is underdetermined, if we leave $C$ unspecified in (80) (or (179)), we see that there are in fact lines of free fixed points parameterised by $C$. The critical exponents along a given line do not vary, therefore we understand that all fixed points appearing on the same line belong to a single universality class.

Let us now relate this to a frame transformation. If we are at a free fixed point of the form

$$\Gamma_\star = C\frac{1}{2}\varphi \cdot \left(-\partial^2\right)^{\frac{1}{2}s} \cdot \varphi, \tag{182}$$

then making the transformation (39) with

$$\epsilon \hat{\xi}[\hat{\chi}] = \frac{1}{2}\hat{\phi}[\hat{\chi}]\,\delta C \tag{183}$$

and using (47), we see that (182) transforms as

$$\Gamma_\star \to C\frac{1}{2}\varphi \cdot \left(-\partial^2\right)^{\frac{1}{2}s} \cdot \varphi + \frac{1}{2}\delta C \varphi \cdot \left(-\partial^2\right)^{\frac{1}{2}s} \cdot \varphi + \text{const}, \tag{184}$$

where the second term comes from the piece proportional to the equation of motion in equation (47), while the constant from the trace term. Thus we obtain a new fixed point where the factor $C \to C + \delta C$ and the vacuum energy is shifted. A change in an inessential coupling at the fixed point is therefore equivalent to a frame transformation that merely moves us along the line of fixed points corresponding to the same universality class.

## D  Calculations

In this Appendix, we specialise the general flow Eq. (56) to the second order in the derivative expansion, explicitly performing the computations needed to retrieve Eqs. (108). In Subsection D.1 we choose to work in momentum space: this part is more suitable to problems characterised by translational invariance for which the calculations are made easier by the availability of the Fourier transform. In Subsection D.2 instead, by taking advantage of the heat kernel formalism, we perform the same computations in position space, as this provides an alternative framework for problems where the translational invariance is lost, like curved spaces and/or boundaries.

### D.1  Momentum space

Hereafter, we adopt the local potential approximation scheme (106). Let's consider the following functional derivatives of the EAA $\Gamma_k$, namely

$$
\Gamma_k^{(2)}(x_1, x_2) \equiv \frac{\delta^2 \Gamma_k}{\delta\phi(x_1)\delta\phi(x_2)} = \int_x \left[ \partial_\mu \delta_{x,x_1} \partial_\mu \delta_{x,x_2} + V_k^{(2)}(\phi(x)) \delta_{x,x_1}\delta_{x,x_1} \right] ,
$$

$$
\frac{\delta\Gamma_k^{(2)}(x_1, x_2)}{\delta\phi(x_3)} = \int_x V_k^{(3)}(\phi(x)) \delta_{x,x_1}\delta_{x,x_2}\delta_{x,x_3} ,
$$

$$
\frac{\delta^2\Gamma_k^{(2)}(x_1, x_2)}{\delta\phi(x_3)\delta\phi(x_4)} = \int_x V_k^{(4)}(\phi(x)) \delta_{x,x_1}\delta_{x,x_2}\delta_{x,x_3}\delta_{x,x_4} ,
$$

(185)

where by $\delta_{x_1,x_2}$ we indicate the $d$-dimensional Dirac delta, i.e. $\delta(x_1-x_2)$. We now consider the Fourier transform of Eq. (185) for a constant field configuration which can be expressed as

$$
\int_{x_1,x_2} \Gamma_k^{(2)}(x_1, x_2) e^{i(p_1 x_1 + p_2 x_2)} = \left( p_1^2 + V_k^{(2)} \right)(2\pi)^d \delta(p_1 + p_2) ,
$$

$$
\int_{x_1,x_2,x_3} \frac{\delta\Gamma_k^{(2)}(x_1, x_2)}{\delta\phi(x_3)} e^{i(p_1 x_1 + p_2 x_2 + p_3 x_3)} = V_k^{(3)} (2\pi)^d \delta(p_1 + p_2 + p_3) ,
$$

$$
\int_{x_1,x_2,x_3,x_4} \frac{\delta^2\Gamma_k^{(2)}(x_1, x_2)}{\delta\phi(x_3)\delta\phi(x_4)} e^{i(p_1 x_1 + p_2 x_2 + p_3 x_3 + p_4 x_4)} = V_k^{(4)} (2\pi)^d \delta(p_1 + p_2 + p_3 + p_4) ,
$$

(186)

where we have suppressed the spacetime indices in order to lighten the notation. In the same way, we can write

$$
\mathcal{R}_k(x_1, x_2) = \int_p \mathcal{R}_k(p) e^{-ip(x_1 - x_2)} ,
$$

(187)

$$
G_k(x_1, x_2) = \left( \Gamma_k^{(2)} + \mathcal{R}_k \right)^{-1}(x_1, x_2) = \int_p G_k(p) e^{-ip(x_1 - x_2)} ,
$$

(188)

$$
G_k(p) = \left( p^2 + \mathcal{R}_k(p) + V_k^{(2)} \right)^{-1} ,
$$

(189)

$$
\frac{\delta}{\delta\phi(x_2)} \Psi_k(x_1) = F_k^{(1)}(\phi(x_1))\delta_{x_1,x_2} = \int_p F_k^{(1)}(\phi(x_1)) e^{-ip(x_1 - x_2)} .
$$

(190)

We notice here that while $G_k$ and $\Psi_k$ are functions of the field, the cutoff function $\mathcal{R}_k$ is not. The l.h.s. of Eq. (56) then reads

$$
\partial_t \Gamma_k + \int_x \frac{\delta\Gamma_k[\phi]}{\delta\phi(x)} F_k(\phi(x)) = \int_x \left[ \partial_t V_k + F_k^{(1)}(\phi)(\partial_\mu \phi)(\partial_\mu \phi) + F_k(\phi) V_k^{(1)}(\phi) \right] ,
$$

(191)

while the r.h.s. of Eq. (56) is composed by two terms, namely

$$\frac{1}{2}\int_{x_1,x_2} G_k(x_1,x_2)\partial_t\mathcal{R}_k(x_2,x_1) = \frac{1}{2}\int_{x_1,x_2}\int_{p_1,p_2} G_k(p_1)\partial_t\mathcal{R}_k(p_2)\mathrm{e}^{-\mathrm{i}p_1(x_1-x_2)-\mathrm{i}p_2(x_2-x_1)}$$
$$= \frac{1}{2}\int_x\int_p G_k(p)\,\partial_t\mathcal{R}_k(p)\,, \tag{192}$$

$$\int_{x_1,x_2,x_3} G_k(x_1,x_2)\frac{\delta}{\delta\phi(x_2)}\Psi_k(x_3)\mathcal{R}_k(x_3,x_1) = \int_{x_1,x_2}\int_{p_1,p_2} G_k(p_1)F_k^{(1)}\mathcal{R}_k(p_2)\mathrm{e}^{-\mathrm{i}p_1(x_1-x_2)-\mathrm{i}p_2(x_2-x_1)}$$
$$= \int_x\int_p G_k(p)F_k^{(1)}\mathcal{R}_k(p)\,. \tag{193}$$

Changing then variables in the remaining momentum integrals as $p \to z = p^2$, the r.h.s. of Eq. (56) can be written as

$$\frac{1}{2}\mathrm{Tr}\frac{1}{\Gamma_k^{(2)}+\mathcal{R}_k}\cdot\left(\partial_t\mathcal{R}_k+2\frac{\delta}{\delta\phi}\Psi_k\cdot\mathcal{R}_k\right) = \frac{1}{2(4\pi)^{d/2}}\int_x Q_{d/2}\left[G_k\left(\partial_t\mathcal{R}_k+2F_k^{(1)}\mathcal{R}_k\right)\right]\,, \tag{194}$$

where the $Q$-functionals are defined in Eq. (111). Considering a constant field configuration and equating (191) and (194) yields the flow equation (108a) for the effective potential $V_k$.

We now take the second derivative of Eq. (56) with respect to $\phi(x)$ and $\phi(\bar{x})$, we impose a constant field configuration and then we Fourier transform, so that the l.h.s. reads

$$\int_{x,\bar{x},x_1}\left\{\delta_{x,x_1}\delta_{\bar{x},x_1}\left[\partial_t V_k^{(2)}\left(\phi(x_1)\right)+\left(F_k\left(\phi(x_1)\right)V_k^{(1)}\left(\phi(x_1)\right)\right)^{(2)}\right]+2F_k^{(1)}(\phi(x_1))\partial_\mu\delta_{x,x_1}\partial_\mu\delta_{\bar{x},x_1}\right\}\mathrm{e}^{\mathrm{i}p_1 x+\mathrm{i}p_2\bar{x}}$$
$$= (2\pi)^d\delta(p_1+p_2)\left[\frac{\delta^2}{\delta\phi(p_1)\delta\phi(-p_1)}\left(\partial_t V_k+F_k V_k^{(1)}\right)+2F_k^{(1)}p_1^2\right]\,. \tag{195}$$

Let's now call $\mathbb{T}$ the trace on the r.h.s. of Eq. (56). Then differentiating w.r.t. $\phi(x)$ and $\phi(\bar{x})$ yields

$$\mathbb{T}_{x\bar{x}} = -\frac{1}{2}\prod_{i=1}^4\int_{x_i} G_k(x_1,x_2)\frac{\delta^2\Gamma_k^{(2)}(x_2,x_3)}{\delta\phi(x)\delta\phi(\bar{x})}G_k(x_3,x_4)\,\partial_t\mathcal{R}_k(x_4,x_1)$$
$$-\prod_{i=1}^5\int_{x_i} G_k(x_1,x_2)\frac{\delta^2\Gamma_k^{(2)}(x_2,x_3)}{\delta\phi(x)\delta\phi(\bar{x})}G_k(x_3,x_4)\frac{\delta\Psi_k(x_5)}{\delta\phi(x_4)}\mathcal{R}_k(x_5,x_1)$$
$$+\frac{1}{2}\prod_{i=1}^6\int_{x_i} G_k(x_1,x_2)\frac{\delta\Gamma_k^{(2)}(x_2,x_3)}{\delta\phi(x)}G_k(x_3,x_4)\frac{\delta\Gamma_k^{(2)}(x_4,x_5)}{\delta\phi(\bar{x})}G_k(x_5,x_6)\,\partial_t\mathcal{R}_k(x_6,x_1)$$
$$+\prod_{i=1}^7\int_{x_i} G_k(x_1,x_2)\frac{\delta\Gamma_k^{(2)}(x_2,x_3)}{\delta\phi(x)}G_k(x_3,x_4)\frac{\delta\Gamma_k^{(2)}(x_4,x_5)}{\delta\phi(\bar{x})}G_k(x_5,x_6)\frac{\delta\Psi_k(x_7)}{\delta\phi(x_6)}\mathcal{R}_k(x_7,x_1)$$
$$+\frac{1}{2}\prod_{i=1}^6\int_{x_i} G_k(x_1,x_2)\frac{\delta\Gamma_k^{(2)}(x_2,x_3)}{\delta\phi(\bar{x})}G_k(x_3,x_4)\frac{\delta\Gamma_k^{(2)}(x_4,x_5)}{\delta\phi(x)}G_k(x_5,x_6)\,\partial_t\mathcal{R}_k(x_6,x_1)$$
$$+\prod_{i=1}^7\int_{x_i} G_k(x_1,x_2)\frac{\delta\Gamma_k^{(2)}(x_2,x_3)}{\delta\phi(\bar{x})}G_k(x_3,x_4)\frac{\delta\Gamma_k^{(2)}(x_4,x_5)}{\delta\phi(x)}G_k(x_5,x_6)\frac{\delta\Psi_k(x_7)}{\delta\phi(x_6)}\mathcal{R}_k(x_7,x_1)$$
$$+\prod_{i=1}^3\int_{x_i} G_k(x_1,x_2)\frac{\delta^3\Psi_k(x_3)}{\delta\phi(x)\delta\phi(\bar{x})\delta\phi(x_2)}\mathcal{R}_k(x_3,x_1)$$
$$-\prod_{i=1}^5\int_{x_i} G_k(x_1,x_2)\frac{\delta\Gamma_k^{(2)}(x_2,x_3)}{\delta\phi(x)}G_k(x_3,x_4)\frac{\delta^2\Psi_k(x_5)}{\delta\phi(\bar{x})\delta\phi(x_4)}\mathcal{R}_k(x_5,x_1)$$
$$-\prod_{i=1}^5\int_{x_i} G_k(x_1,x_2)\frac{\delta\Gamma_k^{(2)}(x_2,x_3)}{\delta\phi(\bar{x})}G_k(x_3,x_4)\frac{\delta^2\Psi_k(x_5)}{\delta\phi(x)\delta\phi(x_4)}\mathcal{R}_k(x_5,x_1)\,. \tag{196}$$

Using equations (185) and (190) and imposing a constant field configuration we have

$$
\begin{aligned}
\mathbb{T}_{x\bar{x}} = & -\frac{1}{2}V_k^{(4)}\delta_{x,\bar{x}}\int_{x_1,x_2} G_k(x_1,x)\,G_k(x,x_2)\left[\partial_t\mathcal{R}_k(x_2,x_1)+2F_k^{(1)}\mathcal{R}_k(x_2,x_1)\right] \\
& +\frac{1}{2}\left(V_k^{(3)}\right)^2\int_{x_1,x_2} G_k(x_1,x)\,G_k(x,\bar{x})\,G_k(\bar{x},x_2)\left[\partial_t\mathcal{R}_k(x_2,x_1)+2F_k^{(1)}\mathcal{R}_k(x_2,x_1)\right] \\
& +\frac{1}{2}\left(V_k^{(3)}\right)^2\int_{x_1,x_2} G_k(x_1,\bar{x})\,G_k(\bar{x},x)\,G_k(x,x_2)\left[\partial_t\mathcal{R}_k(x_2,x_1)+2F_k^{(1)}\mathcal{R}_k(x_2,x_1)\right] \\
& +F_k^{(3)}\delta_{x,\bar{x}}\int_{x_1} G_k(x_1,x)\,\mathcal{R}_k(x,x_1) \\
& -V_k^{(3)}F_k^{(2)}\int_{x_1} G_k(x_1,x)\,G_k(x,\bar{x})\,\mathcal{R}_k(\bar{x},x_1) \\
& -V_k^{(3)}F_k^{(2)}\int_{x_1} G_k(x_1,\bar{x})\,G_k(\bar{x},x)\,\mathcal{R}_k(x,x_1)\,.
\end{aligned}
\tag{197}
$$

Using then equations (188) and (187)

$$
\begin{aligned}
\mathbb{T}_{x\bar{x}} = & -\frac{1}{2}V_k^{(4)}\delta_{x,\bar{x}}\int_{p_1} G_k(p_1)^2\left[\partial_t\mathcal{R}_k(p_1)+2F_k^{(1)}\mathcal{R}_k(p_1)\right] \\
& +\frac{1}{2}\left(V_k^{(3)}\right)^2\int_{p_1,p_2} G_k(p_1)\,G_k(p_2)\,G_k(p_1)\left[\partial_t\mathcal{R}_k(p_1)+2F_k^{(1)}\mathcal{R}_k(p_1)\right]\mathrm{e}^{\mathrm{i}x(p_1-p_2)-\mathrm{i}\bar{x}(p_1-p_2)} \\
& +\frac{1}{2}\left(V_k^{(3)}\right)^2\int_{p_1,p_2} G_k(p_1)\,G_k(p_2)\,G_k(p_1)\left[\partial_t\mathcal{R}_k(p_1)+2F_k^{(1)}\mathcal{R}_k(p_1)\right]\mathrm{e}^{-\mathrm{i}x(p_1-p_2)+\mathrm{i}\bar{x}(p_1-p_2)} \\
& +F_k^{(3)}\delta_{x,\bar{x}}\int_{p_1} G_k(p_1)\,\mathcal{R}_k(p_1) \\
& -V_k^{(3)}F_k^{(2)}\int_{p_1,p_2} G_k(p_1)\,G_k(p_2)\,\mathcal{R}_k(p_1)\mathrm{e}^{\mathrm{i}x(p_1-p_2)-\mathrm{i}\bar{x}(p_1-p_2)} \\
& -V_k^{(3)}F_k^{(2)}\int_{p_1,p_2} G_k(p_1)\,G_k(p_2)\,\mathcal{R}_k(p_1)\mathrm{e}^{-\mathrm{i}x(p_1-p_2)+\mathrm{i}\bar{x}(p_1-p_2)}\,,
\end{aligned}
\tag{198}
$$

and expressing the previous equation in momentum space we obtain

$$
\begin{aligned}
\mathbb{T}_{p_1p_2} = & -\frac{1}{2}V_k^{(4)}(2\pi)^d\delta(p_1+p_2)\int_p G_k(p)^2\left[\partial_t\mathcal{R}_k(p)+2F_k^{(1)}\mathcal{R}_k(p)\right] \\
& +\left(V_k^{(3)}\right)^2(2\pi)^d\delta(p_1+p_2)\int_p G_k(p)\,G_k(p+p_1)\,G_k(p)\left[\partial_t\mathcal{R}_k(p)+2F_k^{(1)}\mathcal{R}_k(p)\right] \\
& +F_k^{(3)}(2\pi)^d\delta(p_1+p_2)\int_p G_k(p)\,\mathcal{R}_k(p) \\
& -2V_k^{(3)}F_k^{(2)}(2\pi)^d\delta(p_1+p_2)\int_p G_k(p)\,G_k(p+p_1)\,\mathcal{R}_k(p)\,.
\end{aligned}
\tag{199}
$$

We then need to expand the previous equation for small $p_1$; for this purpose, we make use of the following expression

$$
f\left((p+p_1)^2\right)=f(p^2)+(p_1^2+2\,p_1\cdot p)f'(p^2)+2\,(p_1\cdot p)^2\,f''(p^2)+O(p_1^3)\,,
\tag{200}
$$

in which primes denote derivatives with respect to $p^2$. Equating then (195) and (199), simplifying a common factor $(2\pi)^d\delta(p_1+p_2)$ on both sides and changing variables as

$p \to z = p^2$ we obtain

$$\frac{\delta^2}{\delta\phi(p_1)\delta\phi(-p_1)}\left(\partial_t V_k^{(2)} + F_k\, V_k^{(1)}\right) + 2F_k^{(1)} p_1^2 = -V_k^{(4)}\frac{1}{2(4\pi)^{d/2}}Q_{d/2}\left[G_k^2\left(\partial_t \mathcal{R}_k + 2F_k^{(1)}\mathcal{R}_k\right)\right] +$$

$$+ F_k^{(3)}\frac{1}{(4\pi)^{d/2}}Q_{d/2}\left[G_k\,\mathcal{R}_k\right] + \frac{\left(V_k^{(3)}\right)^2}{(4\pi)^{d/2}}\left\{Q_{d/2}\left[G_k^3\left(\partial_t \mathcal{R}_k + 2F_k^{(1)}\mathcal{R}_k\right)\right] +\right.$$

$$+ p_1^2 Q_{d/2}\left[G_k'\,G_k^2\left(\partial_t \mathcal{R}_k + 2F_k^{(1)}\mathcal{R}_k\right)\right] + p_1^2 Q_{d/2+1}\left[G_k''\,G_k^2\left(\partial_t \mathcal{R}_k + 2F_k^{(1)}\mathcal{R}_k\right)\right]\right\} +$$

$$- V_k^{(3)} F_k^{(2)}\frac{2}{(4\pi)^{d/2}}\left\{Q_{d/2}\left[G_k^2\,\mathcal{R}_k\right] + p_1^2 Q_{d/2}\left[G_k'\,G_k\,\mathcal{R}_k\right] + p_1^2 Q_{d/2+1}\left[G_k''\,G_k\,\mathcal{R}_k\right]\right\} + O(p_1^4)\,. \tag{201}$$

By finally taking the derivative with respect to $p_1^2$ and then the limit $p_1 \to 0$, we obtain Eq. (108b) .

## D.2  Position space

We revisit the derivation of Eqs. (108), but now working in position space. In order to lighten the notation, we drop the $k$ subscript and leave it intended throughout the whole section. Let's commence by writing the field as

$$\phi(x) \to \phi + \delta\phi(x)\,, \tag{202}$$

where $\phi$ is now understood as constant and if no argument is shown it means that a function of the field is evaluated at $\phi$. Then we write

$$\Gamma^{(2)} + \mathcal{R} = G^{-1} + X\,, \tag{203}$$

where $G^{-1} = -\partial^2 + \mathcal{R} + V^{(2)}$ and we define the following quantities

$$X = V^{(3)}\delta\phi + \frac{1}{2}V^{(4)}\delta\phi^2 + \dots\,, \tag{204}$$

$$\Psi^{(1)} = F^{(1)} + Y\,, \tag{205}$$

$$Y = F^{(2)}\delta\phi + \frac{1}{2}F^{(3)}\delta\phi^2 + \dots\,. \tag{206}$$

The idea now is to expand in $\delta\phi$ and then put the traces into the form $\text{Tr}[\mathcal{O}f(\Delta)]$ and $\text{Tr}[\mathcal{O}^{\mu\nu}\partial_\mu\partial_\nu f(\Delta)]$, where $\mathcal{O}$ are non-derivative operators that might depend on $\delta\phi$ and its derivatives and $f(\Delta)$ is expressed as

$$f(\Delta) = \int_0^\infty \mathrm{d}s\, \tilde{f}(s) H(s,\Delta)\,, \tag{207}$$

where $H(s,\Delta)(x_1,x_2) = \mathrm{e}^{-s\Delta}(x_1,x_2)$ is the heat kernel

$$H(s,\Delta)(x_1,x_2) = \frac{1}{(4\pi s)^{\frac{1}{2}}}\mathrm{e}^{-\frac{1}{4s}(x_1-x_2)\cdot(x_1-x_2)}\,. \tag{208}$$

By taking advantage of the fact that at $x_1 = x_2$, we have

$$H(s,x,x) = \frac{1}{(4\pi s)^{d/2}}\,,$$

$$\partial_\mu\partial_\nu H(s,x,x) = -\frac{\delta_{\mu\nu}}{2(4\pi)^{d/2}s^{d/2+1}}\,, \tag{209}$$

where the derivatives act on the first argument, and therefore one can express the following traces as

$$\text{Tr}[\mathcal{O}f(\Delta)] = \frac{1}{(4\pi)^{d/2}} \int_x \mathcal{O}\, Q_{d/2}[f]\,, \tag{210}$$

$$\text{Tr}[\mathcal{O}^{\mu\nu}\partial_\mu\partial_\nu f(\Delta)] = -\frac{1}{2}\frac{1}{(4\pi)^{d/2}} \int_x \mathcal{O}_{\mu\mu}\, Q_{d/2+1}[f]\,, \tag{211}$$

where

$$Q_n[f] = \int_0^\infty \mathrm{d}s\, s^{-n} \tilde{f}(s) \tag{212}$$

are the equal to the Q-functionals (111). In order to get the flow of the potential $V$, we then want to set $X = 0$ and $Y = 0$. The l.h.s. of the flow equation (56) at constant field is given by

$$\int_x \left[ \partial_t V(\phi) + F(\phi) V^{(1)}(\phi) \right], \tag{213}$$

while the trace appearing on the r.h.s. of equation (56) is given by

$$\frac{1}{2}\text{Tr}[(\partial_t\mathcal{R} + 2F^{(1)}\mathcal{R})G] = \int_0^\infty \mathrm{d}s\, \tilde{W}[(\partial_t\mathcal{R} + 2F^{(1)}\mathcal{R})G, s]\text{Tr}[H(s)]$$
$$= \int_x \frac{1}{2(4\pi)^{d/2}} Q_{d/2}[(\partial_t\mathcal{R} + 2F^{(1)}\mathcal{R})G]\,, \tag{214}$$

where we use the heat kernel expansion to calculate the trace. We therefore retrieve Eq. (108a). By expanding in $\delta\phi$, one we can find the term which involves $\delta\phi\Delta\delta\phi$ on both the l.h.s. and on the r.h.s. of the flow equation (56). On the l.h.s. this yields

$$F^{(1)}(\phi)\,\delta\phi\Delta\delta\phi\,, \tag{215}$$

while on the r.h.s. of the flow equation we obtain

$$\mathbb{T} = \frac{1}{2}\text{Tr}[(\partial_t\mathcal{R} + 2F^{(1)}\mathcal{R} + 2Y\mathcal{R})(G - GXG + GXGXG + ...]$$
$$= \frac{1}{2}\text{Tr}[(\partial_t\mathcal{R} + 2F^{(1)}\mathcal{R})G] - \frac{1}{2}\text{Tr}[XG^2(\partial_t\mathcal{R} + 2F^{(1)}\mathcal{R})] + \text{Tr}[Y\mathcal{R}G]$$
$$+ \frac{1}{2}\text{Tr}[XGXG^2(\partial_t\mathcal{R} + 2F^{(1)}\mathcal{R})] - \text{Tr}[XGY\mathcal{R}G] + \dots. \tag{216}$$

The terms linear in $X$ and $Y$ do not involve derivatives of $\delta\phi$ so we can ignore them. In order to obtain derivatives of $\delta\phi$ we commute $G$ with $X$ and $Y$ which gives the two terms

$$\mathbb{T} \supset \frac{1}{2}\text{Tr}[X[G,X]G^2(\partial_t\mathcal{R} + 2F^{(1)}\mathcal{R})] - \text{Tr}[X[G,Y]\mathcal{R}G]. \tag{217}$$

Then we use $G = G(\Delta)$ where $\Delta = -\partial^2$ to compute the commutators

$$[G,X] \supset -[X,\Delta]G'(\Delta) + \frac{1}{2}[[X,\Delta],\Delta]G''(\Delta)\,, \tag{218}$$

$$[X,\Delta] = X_{,\mu\mu} + 2X_{,\mu}\partial_\mu\,, \tag{219}$$

$$[[X,\Delta],\Delta] = X_{,\mu\mu\nu\nu} + 4X_{,\mu\mu\nu}\partial_\nu + 4X_{,\mu\nu}\partial_\mu\partial_\nu \tag{220}$$

and similarly for $Y$ where the indices after the comma denote derivatives of $X$ with respect to $x^\mu$. The interesting terms are the ones where two derivatives act on $X$ or $Y$. So the

traces we need are

$$
\mathbb{T} \supset \frac{1}{2}\mathrm{Tr}[X(-X_{,\mu\mu}G'(\Delta) + 2X_{,\mu\nu}\partial_\mu\partial_\nu G''(\Delta))G^2(\partial_t\mathcal{R} + 2F^{(1)}\mathcal{R})]+
$$
$$
- \mathrm{Tr}[X(-Y_{,\mu\mu}G'(\Delta) + 2Y_{,\mu\nu}\partial_\mu\partial_\nu G''(\Delta))\mathcal{R}G]
$$
$$
= \frac{1}{(4\pi)^{d/2}} \int_x \left( -\frac{1}{2}XX_{,\mu\mu}\left(Q_{d/2}[G'G^2(\partial_t\mathcal{R} + 2F^{(1)}\mathcal{R})] + Q_{d/2+1}[G''(\partial_t\mathcal{R} + 2F^{(1)}\mathcal{R})]\right) \right.
$$
$$
+ XY_{,\mu\mu}\left(Q_{d/2}[G'\mathcal{R}G] + Q_{d/2+1}[G''\mathcal{R}G]\right) \bigg)
$$
$$
= -\int_x \delta\phi\partial^2\delta\phi\left(\frac{1}{2}\left(V^{(3)}\right)^2\left(Q_{d/2}[G'G^2(\partial_t\mathcal{R} + 2F^{(1)}\mathcal{R})] + Q_{d/2+1}[G''(\partial_t\mathcal{R} + 2F^{(1)}\mathcal{R})]\right)\right.
$$
$$
- V^{(3)}F^{(2)}\left(Q_{d/2}[G'\mathcal{R}G] + Q_{d/2+1}[G''\mathcal{R}G]\right) \bigg) + O(\delta\phi^3), \tag{221}
$$

which upon equating with Eq. (215) completes the derivation of equation (108b).

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
