# Peer review of "Essential renormalisation group"

_SciPost Physics_

## Round 1 · Referee Report · Anonymous · 2021-8-25

Report
The manuscript contains a discussion of field redefinitions in the context of the functional renormalization group formalism. The authors argue that one can redefine fields in a rather general way, and that physical observables should be independent of such changes of coordinates or frame changes in field space. They introduce this as a "principle of of frame invariance in QFT" and argue on this basis that one can distinguish between "essential" and "inessential" operators and couplings in a flowing effective action and that the latter can in fact be eliminated through a choice of frame.
I have the following concerns:
(1) As a general comment, the manuscript is written in an exhaustive and partly excessive and poetic style. This makes it difficult to digest. Moreover, some of the statements made are not supported enough by factual arguments and seem rather to be a product of the imagination of the authors. I would like to strongly encourage the authors to make substantial cuts throughout the manuscript and reduce everything to the, well, essential.
(2) The authors claim that in their scheme the propagator is always almost of the same form as in the microscopic, un-renormalized action (see eq. (93)). This is strange, because one expects on general ground that a full propagator has a much more involved form. The spectral density, in terms of which it can be written through the Källen-Lehmann representation, has a lot of non-trivial physics information, e.g. about the bound state spectrum, resonances, multi-particle branch cuts, scattering continuum etc. pp. Where is all this information if the propagator is just of the microscopic form at all scales?
(3) The UV regularization of the functional integral in eq. (13) is very implicit. How precisely is the theory regularized, and do the authors make sure that this regularization is independent of the non-linear field redefinitions they do later, or is it supposed to change as well? Because this point is left implicit, several places later in the manuscript also remain unclear, for example around eqs. (37) and (58).
(4) The most severe problem with the construction is visible in eq. (57). The right hand side is the expectation value of an infinitesimal change of the field or operator that is coupled to the source $J$ and regulator $R_k$ in the construction of the flowing average action used by the authors. The authors later chose rather freely different forms of this quantity, such as monomials of the field expectation value. It is not clear that this works, however. More specifically, for linear or affine changes of the field there is no problem, because the expectation value of the change is then simply the change of the expectation value. However, for non-linear field redefinitions this is more involved. For example, when the change is quadratic in the field, the expectation value contains a disconnected part corresponding to the product of expectation values, but also the two-point function, which is itself a non-trivial functional of the field (except for the trivial case of a Gaussian theory). In other words, it would be appropriate to investigate at this point what possibilities for $\Psi_k[\phi]$ there are in fact, and not just to assume through an implicit construction that any polynomial can be realized. An implicit construction as the authors pursue it, has two major problems: first it is not clear that it actually really works, and second it is not clear which correlation functions are in the end described by the resulting effective action. In other words, for a given assumed form of $\Psi_k[\phi]$ it is not clear what $\hat\phi_k[\chi]$ actually is!
(5) At this point another question arises. An alternative procedure to do non-linear field redefinitions on the flowing effective action would have been to just take the standard functional RG equation with fixed operators or fields (up to the possibility to do affine field redefinitions which are easy to implement), and to then do a non-linear change of fields on this flow equation. This is a much more explicit construction, and it is always clear what happens, but the drawback might be that the resulting (transformed) flow equation is not of one-loop form any more. Can the authors compare their proposal to this option? Would this not be the way to do things properly? Would the definition of essential and inessential couplings be the same?
(6) The entire concept of essential and inessential couplings remains opaque throughout most of the manuscript. It would be good to provide more examples, also beyond the linear wave function renormalization, and to provide these examples early in the text, before the formal construction is developed. Of course, this would mean to also address properly point (3) raised above. With the current presentation and implicit construction, the reader is left with uneasy feelings.
(7) It would be good to write the figure captions such that they can also be understood (to some degree) independent of the main text. This would mean to explain the notation there, or to reference equations etc. where the notation is introduced.
(8) The notations introduced in eq. (69) are not very clear. Can this be improved and explained further, e.g. with examples?
In summary, I believe that the manuscript in its current form is not suitable for publication. I encourage the authors to make substantial revisions, and to also consider the possibility to concentrate on only part of the material. The parts that are kept should be presented more clearly.

---

## Editorial Decision

unknown